# Utilizing and extending superconducting circuit toolbox to simulate analog quantum gravity

M. A. Javed, D. Kruti, A. Kenawy, T. Herrig, C. Koliofoti, O. Kashuba, and R.-P. Riwar[1]

[1] *Peter Grünberg Institute, Theoretical Nanoelectronics,*
*Forschungszentrum Jülich, D-52425 Jülich, Germany*

There has been considerable effort to simulate quantum gravity features in solid state systems, such as analog black holes or wormholes. However, superconducting circuits have so far received only limited attention in this regard. Moreover, for quantum superpositions of spacetime geometries – a highly contentious notion within the quantum gravity community itself – there currently exist no solid state blueprints. We here show how quantum circuit hardware can implement a variety of classical and quantum spacetime geometries on lattices, by both using established circuit elements and introducing new ones. We demonstrate the possibility of a metric sharply changing within a single lattice point, thus entering a regime where the spacetime curvature itself is trans-Planckian, and the Hawking temperature ill-defined. In fact, our approach suggests that stable, thermal event horizons are incompatible with strictly discrete lattice models. We thus propose to directly probe the evaporation of a wormhole by tracking the accumulation of charge and phase quantum fluctuations over short time scales, which are a robust signature even in the presence of a dissipative environment. Moreover, we present a loop-hole for the typical black/white hole ambiguity in lattice simulations: the existence of exceptional points in the dispersion relation allows for the creation of pure black (or white) hole horizons — at the expense of radically changing the interior wormhole dynamics. Finally, based on multistable Josephson junctions, we introduce the notion of quantum inductors: circuit elements that can be prepared in a superposition of different inductance values. Such inductors realise regions with a quantum superposition of analog spacetime. The entanglement of signals with the quantum spacetime can be probed by a type of delayed-choice experiment.

## I. INTRODUCTION

Unifying quantum mechanics and gravity poses several theoretical [1–3] and experimental challenges [4–14], which motivates studying analogs of relativistic quantum effects in experimentally accessible systems [15]. Several ideas have therefore been proposed for simulating gravitational effects in solid-state systems – for example, holographic ideas borrowed from string theory [16–21], or cosmological particle creation [22–27], or more directly, using Unruh's proposal of "sonic black holes" [28] as an inspiration to simulate black holes in labs. Along these lines, apparent event horizons and the resulting Hawking radiation have been studied on various platforms [29–37], most notably in ultracold atomic gases [38–42].

There remain however a number of open questions. First, so far for all solid state implementations the metric can change only over a finite "healing length". If the curvature at the horizon is too weak, the resulting small Hawking temperature may thus be below the threshold of the system's intrinsic temperature [43] – an obstacle that (at least up to now) seems to have only been overcome for cold atom simulators [41]. Relatedly, systems with an apparent event horizon generically do not have a well-defined ground state, such that any coupling to the environment leads to an instability, making it extremely challenging to distinguish intrinsic radiation due to the horizon (which would actually simulate aspects of a universe with nontrivial spacetime metric) and radiation due to environment induced relaxation. In addition, even in the absence of environment, the closed system can in some platforms be spontaneously unstable [38, 39]. Furthermore, lattice realizations (which ultimately concerns all solid state systems) add a number of interesting but challenging facets, such as a natural (though possibly artifical) resolution of the trans-Planckian problem [44–47], and most notably, the fact that any event horizon has both black and white hole character, since the dispersion relation in a periodic Brillouin zone must necessarily cross zero at least twice [31]. Finally, within the quantum gravity community, there also exists the highly contentious idea of *superpositions* of spacetime geometries, originating from attempts at quantizing the metric tensor [48, 49]. In particular, recent theoretical proposals suggest experiments that examine quantum gravitational effects caused by a superposition of different mass configurations [50–57]. To the best of our knowledge, there currently exist no feasible blueprints for solid state systems capable of simulating quantum superpositions of spacetime geometries.

Superconducting circuits are one of the prime candidates for building large-scale quantum hardware [58–60]. The behaviour of these circuits and their interactions with quantised electromagnetic fields in the microwave range are described by circuit quantum electrodynamics (cQED), a formalism that reduces circuits to lumped elements whose phase and charge are canonically conjugate [61–69]. The toolbox of superconducting circuits includes nonlinear elements such as Josephson junctions and linear elements such as capacitors, inductors, and gyrators – a nonreciprocal multi-port element [70–75]. The same nonreciprocal behaviour also occurs due to topological transitions in the transport degrees of

freedom of multiterminal junctions [76–81]. Over the course of decades, it has been shown that networks of superconducting circuits can give rise to a large number of physical phenomena and imitate various quantum field theories from other domains [34, 82–91]. However, the exploration of analog gravity phenomena has so far been limited to Hawking-like radiation in solitons of Josephson junction arrays [32–35, 92–95] which leave only little control over the shape of the spacetime geometry, or recreation of a limited class of curved spacetimes by means of flux control of SQUID arrays [36, 37].

In this work, we demonstrate that networks of Josephson junctions and gyrators can be used as an engineering tool for simulations of quantum-gravity phenomena with unprecedented tunability capabilities, and generate surprising insights specific to lattice systems. The basic idea is that arrays of these elements result in a (nearly) massless scalar field theory describing a quantum field – the superconducting phases of the charge islands – moving within a spatially varying analog spacetime metric. The latter essentially encompasses the local dispersion relation, i.e., the velocities with which right and left signals move with respect to the laboratory frame.

To create an apparent horizon we need a boundary between the "normal region" (region where the signals can move in both directions) and a wormhole region where the signals only move in one direction, i.e. a region with overtilted dispersion relation. As it turns out such a region can be created with *negative* inductors, which we propose achieve by adapting recent insights in flux quench of regular Josephson junctions [69, 96]. Tunable $0 - \pi$ junctions [97–99] may offer an alternative pathway towards negative inductances.

Already for such classical metric profiles, we make a number of important observations specific to lattice simulations. The exceptional tunability of the individual energy scales in the lattice allows to change the analog metric over only a few lattice sites, which would yield for typical charging and Josephson energies Hawking temperatures in the 100mK up to $1K$ range, thus comfortably exceeding usual cryogenic temperatures. But as we show, this comes at the price of a fundamental impossibility to create stable event horizons for strictly discrete lattice systems. Instead, the system becomes unstable right after the quench, resulting in an immediate evaporation starting out from the event horizon. We here embrace this instability, and show how to realize systems with a change of the spacetime over a *single* lattice point, where the curvature of the metric itself is ill-defined at the horizon (in the style of Refs. [44–46], we refer to the curvature as being "trans-Planckian") resulting in a likewise ill-defined Hawking temperature. We thus create a system where the signatures of wormhole collapse are fully disjoint from any finite curvature effects at the horizon – and in doing so, provide an answer to the question of what happens at an event horizon in the otherwise highly speculative limit of diverging curvature. We show that the instability leads to an accumulation

of charge and phase quantum fluctuations, which we expect to be a highly robust signature with respect to environment-induced dissipation, since the latter *reduces* quantum fluctuations (instead of increasing them).

Moreover, we find an instructive loop-hole to the aforementioned ambiguity of black- versus white hole event horizons in lattice systems. Namely, the inductive coupling can go either via nearest or next-to-nearest neighbour nodes. For the former, the dispersion relation inside the wormhole region exhibits an *exceptional point*, such that the eigenspectrum no longer crosses zero energy twice, but instead takes a detour on the complex plane to satisfy periodicity. We thus realize lattice versions of event horizons which are definitely either black- or white hole, but not both. However, this feature radically changes the dynamics in the interior of the wormhole: instead of an evaporation from the horizons outwards the entire wormhole interior evaporates everywhere immediately after the quench.

Finally, we go beyond the simulation of classical apparent horizons, and demonstrate that the cQED platform suggested here allows for the implementation of *superpositions* of spacetime geometries. This idea draws from the recent development of multistable Josephson junctions [100–102], i.e., junctions whose energy-phase relationship has more than one local minimum. Specifically, the engineering of multiple minimas with *different* local curvature creates a type of *quantum* inductor – an element that can be prepared in a superposition of different inductance values. When creating a chain of maximally entangled quantum inductors, we propose to create a superposition of different spacetime geometries. A classical signal traversing this region propagates in a superposition of different velocities with respect to the lab frame, and thus entangles with the quantum spacetime. The entanglement can be probed by a delayed-choice type of experiment, where a Hadamard gate is applied on the quantum inductor array after traversal of the quantum spacetime region, leading to a specific interference pattern.

In an extended outlook, we show that in particular the idea of quantum superpositions of spacetime geometries harbors a great potential for follow-up studies, where one could endow the spacetime itself with a nonzero Hamiltonian, thus working towards much more complex simulations of analog quantum gravity, where both the field moving in curved analog spacetime and the spacetime itself have a quantum coherent and strongly correlated dynamics. Moreover, while we mostly consider long chains in the main part of this work, as they provide a clean interpretation of the effects in the context of curved spacetime, we are aware that they may be challenging to realize experimentally. In the outlook, we therefore also propose experiments on only few lattice points as proofs of principle, which already contain much of the pertinent phenomenology present in long chains.

This paper is organized as follows, section II is a brief

review of scalar field theories on curved spacetime, and a summary of our main accomplishments of this work. In section III we introduce and study the two circuits that will host the apparent horizons that we wish to investigate, we especially focus on the different effect that negative inductances have on the dispersion relations of these two circuits anticipating the differences between the horizons in them. Section IV is a brief detour, where we expand upon the idea of using flux quench on a Josephson junction to get a negative inductance, and in section V we study and characterize the horizons in the two aforementioned circuits, by looking at the time evolution of quantum fluctuations of the conjugate charge and the phase difference. In section VI we hypothesize on the long time fate of the analog horizons and argue that the effect of this intrinsic evaporation is easily distinguishable from the effect of coupling to the environment. Finally, in section VII we present our proposal for observing quantum superpositions of two spacetime geometries using multistable Josephson junctions.

## II. CURVED SPACE TIME METRICS AND THE CIRCUIT SIMULATOR

### II.1. Brief review of field theories with curved spacetime and stability considerations

To set the stage, let us reiterate basic notions related to field theories with nontrivial spacetime metric.

Within this brief review, questions regarding the stability of the considered systems will emerge. To this end, we begin by providing a number of general statements regarding bosonic Hamiltonians (as we focus on scalar bosonic quantum fields). The quantum systems we consider are all described by a Hermitian Hamiltonian. In accordance with Refs. [38, 39], we point out that hermiticity does *not* guarantee stability of the system, due to the special symplectic structure of the Bogoliubov transformation. In particular, two cases need to be distinguished. It may happen that certain eigenvalues are negative, such that the system has no well-defined many-body ground state. Such systems however still evolve in a stable fashion, unless they are coupled to an environment, which will in general lead to a collapse of the system, as negative energy bosonic states can now be occupied by extracting energy from the bath. In the second case, some eigenenergies may become *complex* (again, this is consistent with Hermitian Hamiltonians, see Refs. [38, 39]). Such systems are spontaneously unstable, as they do not require a coupling to a bath to collapse.

In continuous field theories, the action of a field $\phi$ in a $d + 1$-dimensional spacetime is generally given as

$$S = \frac{1}{2} \int dt d^d x \sqrt{-g} \, g^{\mu\nu} \partial_\mu \phi \, \partial_\nu \phi \, , \qquad (1)$$

where the metric tensor $g^{\mu\nu}$ can in general depend on all coordinates. Within actual general relativity, the

metric tensor of course encompasses the entire causal structure of spacetime. However, it is understood since a long time that a wide variety of nonrelativistic systems (condensed matter and beyond) may be described by an action of the same form, however with typical velocities much below the speed of light. These systems provide an analog spacetime in the sense that $g^{\mu\nu}$ looses its concrete relativistic interpretation.

For simplicity, let us consider $1 + 1$ dimensions (note that our simulator recipe works also for more spatial dimensions, as we point out in a moment), and focus on metrics that depend only on the position coordinate, i.e., metrics that are stationary in the laboratory frame. Here we write the action explicitly in a matrix form as

$$S = \frac{1}{2} \int dt dx \, ( \, \partial_t \phi \ \partial_x \phi \, ) \begin{pmatrix} \frac{1}{u} & \frac{v}{u} \\ \frac{v}{u} & \frac{v^2 - u^2}{u} \end{pmatrix} \begin{pmatrix} \partial_t \phi \\ \partial_x \phi \end{pmatrix} \, , \quad (2)$$

where $u$ and $v$ are two independent functions of $x$. Note that the matrix representing the spacetime metric has determinant $-1$, such that the $\sqrt{-g}$ prefactor is here irrelevant. For actions of this form, it is always possible to make a coordinate transformation [43]

$$y = t - \int^x dx' \frac{1}{u(x') + v(x')} \qquad (3)$$

$$z = t + \int^x dx' \frac{1}{u(x') - v(x')} \, , \qquad (4)$$

to bring the action into the simple form

$$S = \int dy dz \partial_y \phi \, \partial_z \phi \, . \qquad (5)$$

In these coordinates, the classical Euler-Lagrange equation is simply $\partial_y \partial_z \phi = 0$, whose solutions are arbitrary superpositions of functions that depend either only on $y$, or only on $z$. Transforming back to $(t, x)$-coordinates via Eqs. (3) and (4), these two separate solutions correspond to waves propagating locally either with velocity $u + v$ or $u - v$. For constant $u, v$ (and $u > 0$), the theory is readily quantized, yielding the Hamiltonian

$$H = \int dk \omega_k a_k^\dagger a_k \, , \qquad (6)$$

where for bosonic fields, $[a_k, a_{k'}^\dagger] = \delta(k - k')$. The dispersion relation $\omega_k = u|k| + vk$ readily provides the group velocity of quantum excitations, such that the classical $y$ and $z$ type solutions here correspond to $k > 0$ and $k < 0$, respectively. For $|v| < u$, the system describes a regular analog light cone with $H$ having only positive many-body eigenvalues, where $v$ corresponds to a finite tilt of the dispersion relation.

For $|v| > u$ the dispersion relation is overtilted, such that one of the branches (either $k < 0$ or $k > 0$) has now negative energies. Negative energies are meaningful in the following sense. As already pointed out above, the Hamiltonian $H$ here has in principle no more well-defined

ground state – indicating a serious instability. However, the eigenvalues remain real, such that the closed quantum system still evolves in a stable fashion. In particular, with a Galilei transformation [103] by, e.g., $-v$, we can undo the tilt (the dispersion relation goes to $u|k|$) and thus return to a regular Hamiltonian with a well-defined ground state. Consequently, for the closed quantum system, there is nothing special about whether or not a particular branch has negative eigenenergies. Matters are markedly different, once we consider open quantum systems. Here, the Galilei transformation applies to both the system Hamiltonian, as well as to the interaction with the environment. Hence, one cannot change a system from stable to unstable by merely moving with respect to it – thus avoiding what would otherwise be a serious conundrum. However, the inverse may well happen: if we fine tune the parameters of the system such that it implements an overtilted dispersion relation in the laboratory rest frame, the coupling to the environment may indeed be such that the system collapses on very fast time scales (usually given by the typical relaxation rates of the considered quantum hardware).

Returning to spatially varying spacetime metrics, apparent event horizons emerge in this field theory when at a given point $v$ crosses from $|v| < |u|$ to $|v| > |u|$. Consequently, the system is separated at the horizon into two halves, one with a well-defined ground state (regular or no tilt in the dispersion relation) and one with no ground state (overtilted). In principle, one could now expect already the closed quantum system to be unstable as the two halves could exchange energy (which would thus lead to complex eigenvalues). But as is well known [43], this does not happen for this particular system. Even in the presence of such a horizon, the system is still described by regular, real eigenvalues, such that at least for the closed system, the evolution is still stable (though in general not in a well-defined ground state). The reason for this is that on both sides of the horizon, the above coordinate transformation to $(y, z)$ still applies, such that for these coordinates, there still exist two well-defined branches of the wave solutions moving at different speeds relative to the lab frame. At the horizon, the transition from $|v| < |u|$ to $|v| > |u|$ results in one – but only one – of the two coordinate transformations (either $y$ or $z$, depending on the sign of $v$) to be singular. Consequently, the branch without singularity simply moves through the event horizon as if it wasn't there. The other branch (with the singularity) is causally disconnected at the horizon, because wave solutions come to a stand-still. Hence the left and right hand side of this branch do not couple.

Since the system is however not in a ground state, meaningful quantum measurements are best defined by anchoring all observables to a basis with well-defined ground state. This is commonly done [43] by imagining the system to be prepared (in the distant past) in the ground state of a spacetime metric without horizon (e.g., perfectly flat spacetime). When now defining measurements with respect to the new eigenbasis,

Hawking radiation emerges. A frequency decomposition of this radiation reveals that it is of thermal nature, given by the curvature at the horizon – e.g., for constant $u$, the temperature is thus proportional to the energy scale $\partial_x v$, evaluated at the horizon.

Crucially, while for the closed system (or for the actual universe) it therefore seems plausible to have stable spacetime metrics with horizons and a related thermal Hawking radiation, it has to be noted that for simulators coupled to an environment, the above is a highly precarious situation – precisely because of the lack of a well-defined ground state. For instance, while the branch with the singularity may well be exactly separated at the horizon for the closed system, the overtilted part very likely becomes unstable simply due to coupling with the environment (see our above Galilei transformation argument, respectively its inverse). Moreover, it cannot be excluded that a (ever so slightly) nonlocal coupling to an environment effectively couples the two causally separated branches, such that energy can be exchanged across the singularity via environment-assisted tunneling. As we now go on to lattice simulators, stability concerns only pile up.

## II.2. Simulation with circuit networks

Here, we summarize our simulation idea by means of quantum circuits, and indicate the main accomplishments of our work.

As we see in Eq. (2), for a system to emulate a field moving on a nontrivial spacetime, we need interactions that provide the terms $\sim \dot{\phi}^2$, $\sim \dot{\phi}\partial_x\phi$, and $\sim (\partial_x\phi)^2$. Quantum circuit theory [62, 64, 69, 71] readily provides elements that (in a certain continuum limit) provide such interactions. For a superconducting node $j$, a finite capacitance ($C_j$) provides to the Lagrangian $\mathcal{L}$ an energy contribution of the form

$$\mathcal{L} \to \mathcal{L} + \frac{C_j}{2}\left(\frac{\dot{\phi}_j}{2e}\right)^2 , \tag{7}$$

whereas a finite inductance ($L_j$) between two neighbouring nodes $j + 1$ and $j$ provides

$$\mathcal{L} \to \mathcal{L} + \frac{1}{2L_j}\left(\frac{\phi_{j+1} - \phi_j}{2e}\right)^2 . \tag{8}$$

In the continuum limit $\phi_j \to \phi(x)$, we indeed get the terms $\sim \dot{\phi}^2$ and $\sim (\partial_x\phi)^2$, respectively.

However, this alone does not provide a tilt of the dispersion relation. This is where gyrators come into play. As pointed out in Ref. [71], a three-port gyrator, connecting two neighbouring nodes $\phi_{j+1}$ and $\phi_j$ to ground (whose phase is set to zero) can be described by the following contribution to the Lagrangian,

$$\mathcal{L} \to \mathcal{L} + G_j(\phi_j\dot{\phi}_{j+1} - \dot{\phi}_j\phi_{j+1}) , \tag{9}$$

where the parameter $G$ is proportional to the transconductance of the gyrator,

$$2eI_j = \frac{\partial \mathcal{L}}{\partial \dot{\phi}_j} = G_j \dot{\phi}_{j+1} = 2eG_j V_{j+1} \qquad (10)$$

$$2eI_{j+1} = \frac{\partial \mathcal{L}}{\partial \dot{\phi}_{j+1}} = -G_j \dot{\phi}_j = -2eG_j V_j . \qquad (11)$$

The gyrator thus has a typical circular transport behaviour, i.e., a voltage at $j$ ($j+1$) induces a current (with reversed sign) at $j+1$ ($j$). The continuum limit (at least a naive continuum limit, as we discuss in detail below) yields indeed a term of the sought-after form $\sim \dot{\phi} \partial_x \phi$ and thus provides a tilt. In fact, it is precisely the above circular transport behaviour, which allows us to develop an illustrative intuitive picture for the physical origin of the tilt. Essentially, the gyrator acts as a "conveyer belt" for plasmon excitations within the chain, transporting signals in opposite directions with different group velocity. Thus it has in essence the same effect as a Galilei transformation (at least in the continuum limit), except that we can in principle control it locally within a given chain by modulating the value of $G$ as a function of $j$.

The above ingredient list is in principle sufficient to simulate analog spacetime geometries with finite curvature. Also, while 1D and 2D arrays are likely most straightforwardly implemented on a circuit board, with appropriate 3D stacking one could potentially also reach a higher number of spatial dimensions. The above realizations mark a first central accomplishment of this work, where

(i) we identify a minimal set of circuit elements to realize scalar discrete field theories with any shape of stationary analog spacetime geometry.

For illustration purposes and practicality, we stick to 1D arrays in the remainder of this work. However, even in the $1+1$ dimensional problem, there are a number of interesting challenges which we attack in the following. As we argue in more detail below, already for flat spacetime, an overtilt (with a negative energy branch) is not straightforward to realize. As can be seen when comparing Eq. (2) with the inductor contribution of Eq. (8) an overtilt $|v| > u$ requires *negative* inductances. As a second main result (developed in the subsequent section),

(ii) we show how to realize negative inductances via transient flux drive of Josephson junctions, and thus create overtilted dispersion relations and analog (apparent) event horizons.

This particular implementation of negative inductances via a flux quench is first of all of fundamental interest, as (contrary to other realizations [38, 42]) we do not require a continuous non-equilibrium drive to maintain the event horizons. As we detail further below, the absence of a well-defined ground state is merely the result of an approximation valid at short times. The exact quantum field theory retains a ground state at all times, which will ultimately be reached again via relaxation processes, thus potentially offering the possibility of a complete wormhole evaporation simulation emerging naturally from the hardware. Moreover, the setup we propose allows for a radical change of the spacetime geometry over only a few lattice sites (and as we show in a moment, even over a *single* lattice bond). With the energy scales of the local charging energy given by $E_C$ and when using Josephson junctions as inductive elements $E_J$, we would thus (at least in principle) be able to reach Hawking temperatures (as defined above) on the order of $\sim \sqrt{E_C E_J}$ which (taking as a typical transmon qubit frequency the value of 1GHz $\sim$ 10GHz) can yield up to 0.1K $\sim$ 1K (which is above typical cryogenic temperatures).

However, in addition to the already mentioned general stability issues related to the environment, the circuit simulation reveals fundamental lattice-specific spontaneous instabilities (due to complex eigenvalues), which are already present when disregarding the environment. We expect those to impede a straightforward observation of thermal Hawking radiation. Instead,

(iii) we propose to directly probe the spontaneous collapse of systems with apparent event horizons by means of accumulating quantum fluctuations, which we argue to be distinguishable from collapse due to dissipative processes.

At any rate, some of the lattice-specific stability issues can be appreciated on the general level here, others will be shown subsequently for a specific model. On a general footing, we see in Eq. (2) that any (constant or spatially dependent) local tilt $v$ (which in our simulation would be implemented by constant or spatially varying gyrator parameters $G_j$) must equally appear in the $(\partial_x \phi)^2$ term, as the corresponding matrix element $g^{11}$ ($\mu, \nu = 1$ is the space-space component) is proportional to $v^2 - u^2$. In other words, a finite nonreciprocity in the circuit needs to be "countered" by a corresponding inductive interaction, in order to realize actions of the form of Eq. (2). Consequently, we would need an according spatially dependent tuning of the local inductance profile $L_j$ which matches exactly with the values of $G_j$. But first of all, such a perfect fine tuning is realistically not possible, such that the condition $g = -1$ is in general only approximately, but never exactly, fulfilled. Hence, the mapping onto Eq. (5) is likewise not exact, and we cannot guarantee a perfect causal decoupling across the horizon, and the closed system might become spontaneously unstable over already short time scales. In fact, below we refrain from a simulation with constant metric determinant $g$ (in fact, we will modify the inductances rather than the gyrator values), such that the systems we consider do not map to Eq. (5) in the continuum limit. Furthermore, with the Hawking temperature being defined (at least in continuous field theories) as the derivative of the metric (i.e., the curvature) at the event horizon, there emerges the curious question of what happens to the observables of the system when considering a very sharp (discontinuous) change of

the metric at the horizon, and how such a diverging curvature is regularized. In what follows we will provide an answer to that question for lattice simulators.

Moreover, there is a deeper issue that our discrete circuit realization unravels, which is true even if perfect tuning of $G_j$ and $L_j$ would be available. To see this, consider first an (infinite) array where all gyrators have the same parameter $G_j = G$. Here, we can reformulate the nonreciprocal part of the Lagrangian as follows,

$$G \sum_j \left( \phi_j \dot{\phi}_{j+1} - \phi_{j+1} \dot{\phi}_j \right) = G \sum_j \dot{\phi}_j \left( \phi_{j-1} - \phi_{j+1} \right) .$$

(12)

Thus, we see that in order to have the correct counter term in the $g^{11}$ component for the lattice, we actually do *not* need nearest neighbour inductive coupling [as indicated in Eq. (8)] but instead *next-to-nearest* neighbour coupling, $\sim (\phi_{j+1} - \phi_{j-1})^2$. As further detailed below, because of this fundamental feature of the gyrator element, nearest and next-to-nearest neighbour inductive couplings have fundamentally different stability properties. In particular,

(iv) we find an important loop hole to the ambiguous black and white-hole nature of lattice event horizons (through an exceptional point in the dispersion relation) leading to a fundamentally different behaviour in the wormhole interior.

Finally, when returning to a spatially varying $G_j$, we can draw another important conclusion. Namely, for a general gyrator Lagrangian of the form

$$\sum_j G_j \left( \phi_j \dot{\phi}_{j+1} - \phi_{j+1} \dot{\phi}_j \right) =$$
$$\sum_j \dot{\phi}_j \left( G_{j-1} \phi_{j-1} - G_j \phi_{j+1} \right) ,$$

(13)

it turns out to be impossible to find exact inductive counter terms: as can be seen in the right-hand side of above equation, the Lagrangian can no longer be factored into simple phase differences (due to $G_j \neq G_{j-1}$). The only possibility to counter the above expression in terms of inductive couplings is to add inductors to ground, and thus abandon charge conservation within the array. While charge leakage to ground is very much a potential reality (especially if the realization of the gyrators is not ideal, leading to additional inductive shunts), nonetheless

(v) we conclude that for generic circuits networks, non-reciprocal interactions cannot be exactly countered by charge conserving inductive interactions.

The breaking of charge conservation (which is physical as it corresponds to leakage to ground) effectively provides a mass term in the theory. In fact, we will include a (finite but very small) mass term in the following, but for a different reason: it allows us to deal with the otherwise problematic zero mode in the boson Hamiltonian.

To summarize, items (i) and (v) have already been fully demonstrated within the above general reasoning, and require no further illustration. Items (ii-iv) on the other hand are now explicitly shown in what follows, by considering a concrete setup.

## III. SETUP

We here show with two example circuit models the capability of circuit arrays to engineer nontrivial dispersion relations with analog wormholes. The first proposed circuit consists of an array of $J$ nodes (with periodic boundary conditions, i.e., $J + 1 = 1$), interconnected via Josephson junctions in parallel with 3-port gyrators, as depicted in Fig. 1a). The complete Lagrangian of the circuit will consist of three parts, namely a capacitive Lagrangian in Eq. (7), a gyrator Lagrangian in Eq. (9) and a term for the Josephson junctions $\mathcal{L}_J = \sum_{j=1}^J E_J \cos(\phi_{j+1} - \phi_j + \phi_{\text{ext},j})$. Here, $E_J$ is the Josephson energy and $\phi_{\text{ext},j}$ is the external flux coupling to each Josephson junction. This flux, and its time-dependent control, will play a pivotal role in what follows.

By means of standard circuit theory methods [62, 64], we find that the Hamiltonian of this array takes the form

$$\mathcal{H} = \sum_{j=1}^J E_{\text{C}} \left[ N_j + G(\phi_{j+1} - \phi_{j-1}) \right]^2$$
$$- \sum_{j=1}^J E_J \cos(\phi_{j+1} - \phi_j + \phi_{\text{ext},j}),$$

(14)

with the charging energy

$$E_{\text{C}} \equiv \frac{(2e)^2}{2C} .$$

The number operator $N_j$ and the phase $\phi_j$ satisfy the canonical commutator $[N_j, \phi_{j'}] = i\delta_{jj'}$.

As stated in the previous section, the stability of circuits where Josephson junctions connect nearest neighbours nodes and the ones where Josephson junctions couple next nearest neighbour nodes is very different. Therefore, to illustrate their differences the second circuit we study has next nearest neighbour connections via Josephson junctions, see Fig. 1b). The Hamiltonian for such a circuit is

$$\mathcal{H}' = \sum_{j=1}^J E_{\text{C}} \left[ N_j + G(\phi_{j+1} - \phi_{j-1}) \right]^2$$
$$- \sum_{j=1}^J E'_J \cos(\phi_{j+1} - \phi_{j-1} + \phi_{\text{ext},j}) .$$

(15)

Considering the system at low energies (close to the many-body ground state), the phases on the nodes will approach an equilibrium configuration minimizing the total Josephson junction energy of the array. For

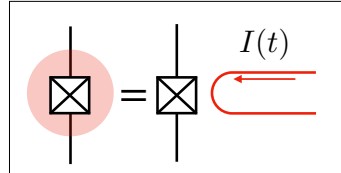

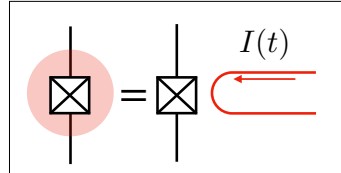

FIG. 1. Wormhole simulations with quantum circuits. In circuits a) and b) (nearest and next-to-nearest Josephson coupling, respectively) the presence of gyrators tilts the dispersion relation giving left and right moving wave packets different speeds. The circuit c) is obtained by inverting the signs of a finite connected region inductances of circuit a). This is achieved by a flux quench induced by a localized current (inset), shifting the superconducting phase difference across the junction by $\pi$, leading to negative inductance. This creates two boundaries between the normal and over-tilted regions, which act as the apparent horizons. Horizons for the circuit with Josephson junctions connecting next-nearest neighbors b) can be obtained in a similar manner.

sufficiently long arrays, we find either $\phi_{j+1} - \phi_j \approx -\phi_{\text{ext},j} + \delta\phi_{j+1} - \delta\phi_j$ (for $\mathcal{H}$) or $\phi_{j+1} - \phi_{j-1} \approx -\phi_{\text{ext},j} + \delta\phi_{j+1} - \delta\phi_{j-1}$ (for $\mathcal{H}'$). Assuming $E_C < E_J$, the quantum fluctuations around the equilibrium value will be small, $\delta\phi_j \ll 1$. Consequently, the Josephson term in Hamiltonians (14) and (15) can be approximated by

$$\approx \sum_{j=1}^{J} E_L (\delta\phi_{j+1} - \delta\phi_j)^2 + \text{const.}$$

$$\approx \sum_{j=1}^{J} E_L' (\delta\phi_{j+1} - \delta\phi_{j-1})^2 + \text{const.}, \qquad (16)$$

where $\delta\phi_j$ still satisfy the same commutation relations with $N_j$. The inductive energy is $E_L^{(\prime)} = E_J^{(\prime)}/2$.

With the quadradic approximation of the Josephson energies, the total Hamiltonian describes a noninteracting

boson field. Moreover, close to the ground state, the externally applied flux $\phi_{\mathrm{ext},j}$ could be eliminated from the problem. It would therefore seem that neither the nonlinearity of the Josephson junction, nor the external flux play a role. However, as we will show in Sec. IV, transient flux-control and the compact cosine behaviour of the Josephson energy will allow us to generate nontrivial features related to quantum gravity. In particular, we will be able to modulate the effective inductive energy $E_L$ highly locally, and in particular, render it negative, allowing for the creation of instabilities that will lead to non-thermal Hawking radiation. Note that the nonlinearity of the Josephson energy was likewise of importance for earlier proposals to measure Hawking-like radiation emerging from soliton excitations in Josephson junction arrays [33]. We will comment on similarities with – but also significant differences to – our approach further below.

For the above conventional non-interacting boson problem, translational invariance allows us to analytically diagonalize the Hamiltonians (14) and (15), $\mathcal{H}^{(\prime)} = \sum_m \omega_m^{(\prime)} b_m^\dagger b_m$ (where index $m$ goes from $-J/2+1$ to $J/2$), to get the dispersion relations (Appendix A)

$$\omega_m = 2\sqrt{E_C^2 \beta_m + 4E_C G \sin\left(\frac{2\pi m}{J}\right)},$$
$$\omega_m' = 2\sqrt{E_C^2 \beta_m' + 4E_C G \sin\left(\frac{2\pi m}{J}\right)}, \quad (17)$$

where $b_m$ $(b_m^\dagger)$ are bosonic annihilation (creation) operators satifying $[b_m, b_{m'}^\dagger] = \delta_{mm'}$ and

$$\beta_m = \frac{4E_C G^2 \sin^2\left(\frac{2\pi m}{J}\right) + 4E_L \sin^2\left(\frac{\pi m}{J}\right) + M}{E_C},$$
$$\beta_m' = \frac{4E_C G^2 \sin^2\left(\frac{2\pi m}{J}\right) + 4E_L' \sin^2\left(\frac{2\pi m}{J}\right) + M}{E_C}, \quad (18)$$

where we will set $E_L' = E_L/4$ for the ease of comparing the two spectra in the linear approximation. The quantum number $m$ enumerates the momentum of bosonic excitations, where we can define the wave vector as $k = 2\pi m/J$, such that $\omega_m^{(\prime)} \to w^{(\prime)}(k)$. Note that we have introduced a very small mass term $M$ ($\mathcal{H}^{(\prime)} \to \mathcal{H}^{(\prime)} + M \sum_j \phi_j^2$) to avoid the well-known diagonalization issues with the zero modes. We take whenever possible the limit $M \to 0$. The mass term physically corresponds to a small leakage of supercurrent towards ground, which could, e.g., originate from an imperfect implementation of the gyrator element.

For small $k$, the system exhibits a linear dispersion relation, [see Figs. 2a) and 2b)],

$$\omega(k) \approx \omega'(k) \approx 2\sqrt{4E_C^2 G^2 + E_C E_L}\,|k| + 4E_C G k. \quad (19)$$

In the relativistic sense, this would correspond to the analog speed of light, where the two branches $k > 0$ and $k < 0$ denote right and left moving signals, respectively

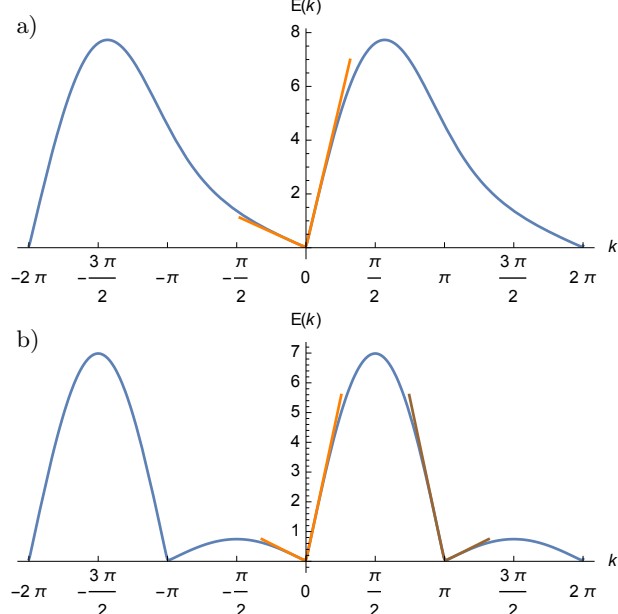

FIG. 2. The above two figures show that the low-momentum eigenvalues (blue lines) of the circuit Hamiltonians (14) and (15) can be approximated by the linear dispersion relation (19) (orange lines) at low energies. The branches of the dispersion relation correspond to right and left movers with speeds $u \pm v$. One crucial point about the eigenvalues in b) is that near $k = \pi$ the dispersion relation (brown lines) is the mirror image of the dispersion near $k = 0$, which is a feature that the eigenvalues in a) do not have. Parameters used: $E_C/E_L = 1.3, E_L'/E_L = 0.25, G = 0.6$, and $J = 50$. To lift the degeneracy at zero energy, a small mass term is used ($\approx 10^{-3} E_L$).

[see also Eq. (6)]. Importantly, we see in general different analog speeds of light $u + v$ $(u - v)$ for excitations moving to the left, $k < 0$ (right, $k > 0$) [as illustrated in Figure 1a) and b)], with $u \propto 2\sqrt{4E_C^2 G^2 + E_C E_L}$ and $v \propto 4E_C G$. A nonzero $v$ corresponds to a tilt of the dispersion relation, which is only possible for a nonzero gyrator, $G \neq 0$, due to the aforementioned "conveyer belt" dynamics.

Moreover, we observe that on this coarse-grained level (small $k$ essentially corresponds to taking the continuum limit of the lattice grid), the two models exhibit the exact same dispersion relation. Note however, that the next-nearest neighbour model hosts a second low-energy point with linear dispersion relation at $k \approx \pi$ [see Fig. 2b)]. The dispersion tilt here is always opposite to the one at $k \approx 0$. This feature emerges for the following reason. In the absence of the gyrators, the model $\mathcal{H}'$ has two disjoint low-energy fields for even and odd lattice sites, resulting in a $\pi$-periodic Brillouin zone. The gyrators couple these two separate modes, restoring $2\pi$-periodicity with respect to $k$ (due to the alternating tilt), but keeping the two low-energy solutions intact. In essence, while model $\mathcal{H}$ hosts a single low-energy analog light field, model $\mathcal{H}'$ hosts two separate fields with opposite dispersion relation. This detail will play an even more prominent role in what follows.

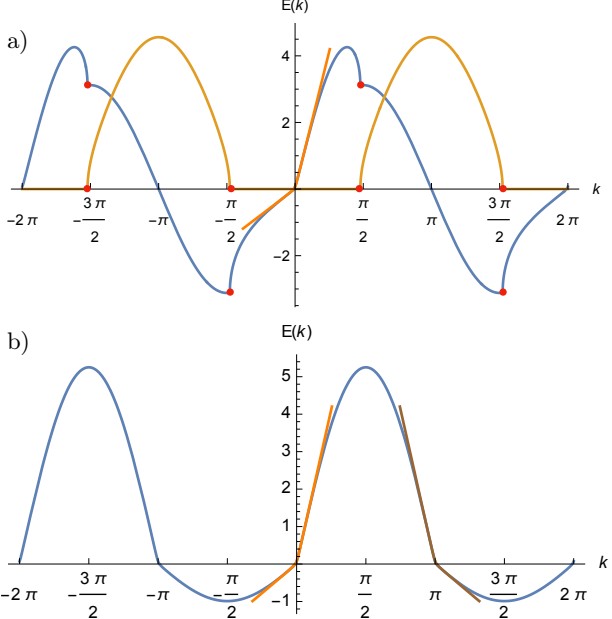

FIG. 3. The principle behind the topological loop-hole for black versus white hole horizons in lattice systems via exceptional point. In panels a) and b), we show the complete dispersion of the circuit Hamiltonians (14) and (15) with negative inductances. When the Josephson junctions connect nearest neighbors, a), the eigenspectrum is complex (blue line represents real part of the eigenvalues and yellow line the imaginary part). The points where the eigenvalues go from real to complex are exceptional points (marked by red dots). Consequently, the spectrum only crosses zero once at $k = 0$, and satisfies the periodicity constraint of the Brillouin zone via a detour in the complex plane. For next nearest neighbor coupling, b), the spectrum is real for all values of $k$, and crosses zero twice, again with mirror images at $k = 0$ and $k = \pi$. The low momentum eigenvalues (orange line) for both the circuits do show an overtilt, which is required to create a horizon. Parameters are same as in Fig. 2, except the signs for $E_L$ and $E_L'$ are flipped.

In order to engineer analog wormholes with event horizons, we need to specifically create a dispersion relation with an *overtilt* where $|v| > u$ – at least in a finite region of the device. Here, our resulting dispersion relation, Eq. (19), provides us with a significant challenge: if both the inductive and capacitive energies $E_L, E_C$ are positive, then it can be easily shown that $u > |v|$, independent of the specific parameter values. Consequently, an overtilt must necessarily involve negative capacitances or negative inductances. In the next section, we will explicitly discuss possibilities to render either $E_C$ or $E_L$ negative, and point out in particular, why we expect negative $E_L$ to be the more feasible of the two options.

In the remainder of this section, we proceed by discussing the consequences a negative inductance has on the dispersion relation. As already stated, for small $k$ we thus get an overtilted linear dispersion relation in both models, see Figs. 3a) and 3b). As the slope of the dispersion relation now has the same sign for both positive and negative $k$, we get signals propagating in the same direction. If we now join regions with regular ($u > |v|$) and overtilted ($u < |v|$) dispersion relation, we could indeed engineer analog wormholes. But before we embark on that, we need to discuss in detail the behaviour of the dispersion relations, Eq. (17), for large $k$. Namely, when considering the full Brillouin zone (i.e., for $-\pi < k < \pi$), we see that the two models (nearest versus next-nearest neighbour) differ significantly. In particular, for the nearest neighbor coupling, we observe that outside the range $-k_0 < k < k_0$ the dispersion relation becomes complex, see Fig. 3a), where $k_0$ is determined by the condition

$$\sin^2\left(\frac{k_0}{2}\right) = \frac{4E_C G^2 - |E_L|}{4E_C G^2}. \qquad (20)$$

This transition point is equivalent to an *exceptional point* in the system. Crucially, this finding is in stark contrast to the model circuit where the Josephson junctions connect the next nearest neighbors. Here, for parameter regime $4E_C G^2 + E_L > 0$, we can overtilt the dispersion relation without encountering complex eigenvalues, see Fig. 3b).

There are several reasons why both of these models are interesting in their own right. First, let us comment in more detail on stability. In principle, the inversion of inductance (or capacitance for that matter) indicates a spontaneous electrostatic instability. This can be understood already on a much simpler level of a single $LC$ circuit, with a resonance frequency $\sim \sqrt{E_L E_C}$. Swapping the sign of the inductance leads to an imaginary frequency. Note that for transient times, this result is by no means problematic (see also Sec. II.1), as such an imaginary frequency corresponds to placing a localized wave function on the tip of an inverted harmonic potential. For transient times, we can by all means obtain physically meaningful results from the Schrödinger equation. In particular, note that such an evolution will result in an exponential blowing up of quantum fluctuations of both canonically conjugate quantities, here, charge and phase (as we will see later also in explicit calculations). This might at first sight seem counterintuitive. The most common situation for generic quantum wave functions is that the uncertainty in one of the two conjugate spaces is inversely proportional to the uncertainty in the other. But note that inverted potentials provide very special wave functions over time that spread both in position (e.g., $\phi$) space, as the wave function evolves from the tip of the parabola to both sides, as well as increasing uncertainty in momentum (e.g., $N$) space, since the particle wave gets more and more accelerated as it rolls down both sides of the inverted parabola. The theory is only problematic in the long-time limit, due to the formal removal of a well-defined ground state. We note that our realization of negative inductances via the inherent nonlinearity of the Josephson effect provides a very neat conceptual realization of instabilities, while (in principle) retaining a well-defined ground state for long times, as we explain in more detail further below. We also note that a recent work [104] has explored a related but distinct concept of negative mass resonators in cQED architectures, by

strongly driving a weakly nonlinear superconducting LC circuit [105], leading effectively to negative susceptibility.

To proceed, we note that in the 1D chain models we consider, there are ways to stabilize the system in spite of negative inductances, in line with the discussion in Sec. II. Indeed, the gyrators play the pivotal role for stabilization: in the Hamiltonians of Eqs. (14) and (15), when we expand the term in the first line $\sim E_C$, we get an effective inductive element due to a nonzero $G$, $\sim E_C G^2 (\phi_{j+1} - \phi_{j-1})^2$. This is an effective inductive contribution due to the gyrator, which is guaranteed to be positive even when $E_L < 0$ (as long as $E_C > 0$). But as already foreshadowed in Sec. II.2, this effective inductive contribution couples next-nearest neighbour sites. Hence, a negative next-nearest inductor [Eq. (14)] can be exactly compensated by a positive next-nearest neighbour contribution of the gyrator. The negative nearest neighbour inductor model [Eq. (15)] on the other hand cannot exactly be compensated by the gyrators, hence the presence of the exceptional point.

Let us now take a bit of a more formal perspective on the above observation for generic wormhole simulations, which will allow us to demonstrate one of the central results of this work, see also item (iv) in Sec. II.2. Namely, there is a simple topological connection between overtilted spectra, Brillouin zone sizes, and exceptional points. For discrete lattice implementations the dispersion relation is always $2\pi$ periodic in $k$. Crucially, an overtilted spectrum means that the energies cross from positive to negative values at the transition points $k = 0, 2\pi, \dots$. If the spectrum is real for all $k$, then periodicity in momentum space always guarantees that an overtilted spectrum near a certain value of $k$ must have an overtilted partner at another value of $k$ with *opposite tilt direction*, as a periodic spectrum must traverse the zero-energy line an even number of times. Consequently, for any lattice implementation, it seems unavoidable that what appears like a black hole horizon near one value of $k$, must necessarily have a white hole horizon partner at another value of $k$. This fact has already been pointed out in the context of analog black-hole simulations in tilted Weyl semimetal structures [31]. Our next-nearest neighbour model confirms this expectation, where the overtilt point at $k \approx 0$ has an inverted partner point at $k \approx \pm\pi$, see Fig. 3 b). These two $k$ points can be individually addressed when preparing, e.g., Gaussian wave packets centered either around $k \approx 0$ (wave packets that are smooth in $j$ space) and $k \approx \pm\pi$ (wave packets with alternating even-odd signs in $j$ space).

Crucially, the nearest-neighbour model provides an intriguing loop hole: here, the overtilt at $k \approx 0$ has no partner point with inverted tilt, very simply because the dispersion relation satisfies the periodicity constraint in $k$ by taking, in a sense, a "detour" in the complex plane – by courtesy of the exceptional point. Consequently, the spectrum crosses the zero energy line only once. We are thus able to conclude that there is after all a case, where black and white hole horizons can be created

independently in a lattice. This comes at the cost of a spontaneous instability within the overtilted region, which, importantly, is present even in the absence of an event horizon. This finding will be of great importance when interpreting the origin of radiation in the presence of event horizons in Sec. V below.

## IV. REALIZING NEGATIVE INDUCTANCE THROUGH QUENCH

As noted above, an overtilt in the dispersion relation is realized by means of either negative capacitances or negative inductances. We here present ideas for the engineering of both features, and then argue, why we expect the latter to be more feasible, thus demonstrating claim (ii) in Sec. II.2,

Negative (and other nonlinear) capacitances are an interesting topic within quantum circuits, which date back to various pioneering ideas by Little [106] and Landauer [107], and have recently been a focal point of research, either in the form of ferroelectric materials [108–111], or capacitively-coupled polarizers inducing pairing in quantum dots [112, 113]. Specifically within the circuit QED context, it has recently been proposed that negative capacitances exist in the sense of an emf-induced renormalization [69], or that the charge in the quantum phase slip energy contribution can be fractionalized and thus rendered incommensurate [114, 115]. There is, however, the problem that negative capacitances correspond to an electrostatic instability, and are thus only meaningful either as partial capacitances (where another capacitance in parallel guarantees total positivity on a given charge island) or in a transient regime. Since we are interested in the latter (after all, Hawking radiation is manifestly a transient feature), this means that the capacitance should have to be tunable on very fast time scales, which seems challenging, especially on the small scales of the here proposed circuit QED architecture.

This is why we move on to ideas on how to bring about a transient behaviour within the inductive (Josephson junction) element. Remember that we focus on a regime of $E_J > E_C$ where the cosine of the Josephson energy can be approximated as a parabola with the inductive energy $E_L \sim E_J$ of an effectively regular linear inductor (rendering the resulting plasmon field effectively non-interacting, as shown above). Nonetheless, we can exploit the nonlinear inductive behaviour. Suppose there is a mechanism, allowing for a fast, transient switch from $-\cos(\phi_j - \phi_{j-1})$ (e.g., for the nearest neighbor model) to $+\cos(\phi_j - \phi_{j-1})$. If that switch occurs sufficiently fast, the circuit degrees of freedom have no time to adjust, such that the many-body wave function immediately after the switch remains localized on what is now the *maximum* (and not the minimum) of the cosine. Expanding for small phase differences localized around said maximum, we can realize a sign switch of the inductive energy, $E_L \to -E_L$, leading to the sought-after overtilt in the dispersion

relation (see the orange lines in Fig. 2). Specifically, in the next section, we discuss the possibility of applying an inverted inductance only in part of the system to create event horizons separating regions with $u < |v|$ from unquenched regions with $u > |v|$.

Our proposal, thus, requires a mechanism to locally address the Josephson junctions in a way to create connected regions with inverted inductance. We note that so-called tunable $0 - \pi$ junctions are a subject of ongoing research and have multiple proposals for their implementation, including the theoretical proposal about Josephson junctions with a high-spin magnetic impurity sandwiched between two superconductors [97], experimental realization in ballistic Dirac semimetal Josephson junctions [98] and another experimental realization in Indium antimonide (InSb) two dimensional electron gases [99].

We here point out a feasible alternative, where inductance inversion is also possible for regular superconductor-insulator-superconductor Josephson junctions. Namely, a nearby current source can likewise induce a phase shift of $\pi$. This idea is based on the results of Ref. [69] where a gauge invariant formulation of time-dependent flux driving was derived. Gauge aspects are of utmost importance to understand how time-dependent flux is allocated in circuits involving multiple Josephson junctions, such as SQUIDs or junction arrays [69, 96, 116]. Previous to Ref. [69], it was expected that the ordinary lumped element approach to circuit QED is valid in the presence of time-dependent flux. In this standard framework, it was recently shown [96] that the allocation of the time-dependent flux within the device (and thus, of the induced electric fields) is given by the capacitive network of the involved charge islands. If this observation were generally true, it would be difficult to locally address certain junctions with high local precision, unless the capacitive network would be engineered accordingly. However, as shown in Ref. [69], the connection between flux allocation and the capacitive network is valid only in special cases; in general, both device geometry and magnetic field distribution lead to a highly nontrivial flux allocation.

We here adopt this principle to show that the individual junctions can be addressed by small current loops, (one loop per junction) where a time-dependent drive (inset in Fig. 1) provides a shift inside the Josephson energies of the form $\cos(\phi_{j+1} - \phi_j + \phi_j^{\text{ext}})$, where

$$\phi_j^{\text{ext}} = \frac{2\pi}{\Phi_0} \int_{\mathcal{L}_j} d\mathbf{l} \cdot \mathbf{A} \ . \qquad (21)$$

Here $\Phi_0$ is the flux quantum and $\mathbf{A}$ is the vector potential in the irrotational gauge [69, 96] accounting for the magnetic field emitted by the current carrying loop sources. Note that since the magnetic field decays only as a power law, one may have to take into account non-local effects, that is, the current loop at junction $j$ influences not only the phase drop at junction $j$ but also at the other junctions $j'$, see Fig. 4a). We can generically write

a)
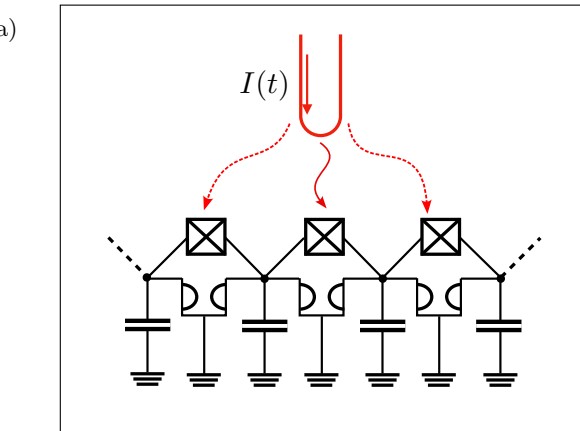

b)
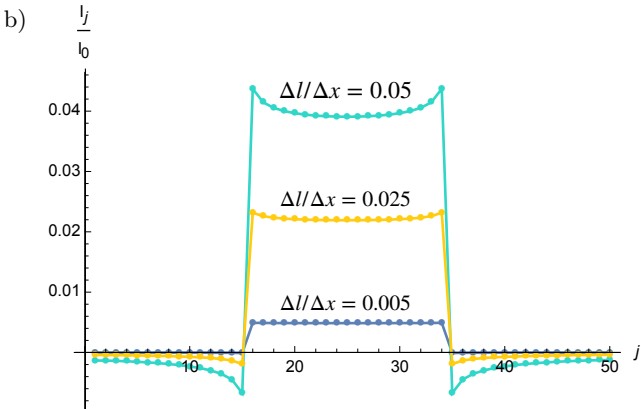

FIG. 4. Illustration of non-local effects of the current loop flux drive. In order to create the target phase shift profile of exactly $\pi$ within a connected region in the junction array (from $j = j_0$ to $j = j_1$), the corresponding applied current profile as a function of site index $j$ needs to be slightly nonlocal. We have plotted three current profiles, that were obtained for a system with 50 sites with $j_0 = 15$ and $j_1 = 35$, with different ratios of $\Delta l$ to $\Delta x$ as shown in the graph. The parameter $R$ does not have a significant effect on the nature of the current profile, only on the magnitude of the ration $I_j/I_0$, hence we set $R/\Delta x = 1$.

down the equation

$$\phi_j^{\text{ext}}(t) = \sum_{j=1}^{J} \alpha_{j,j'} I_j(t) \ , \qquad (22)$$

where the matrix $\alpha$ contains the information about the geometric details of the circuit, and accounts for the screening of the magnetic field (Meissner effect) as well as the screening of the induced electric field (Thomas-Fermi screening). In order to know what exact current pulses need to be applied in order to create a desired target phase profile $\phi_j^{\text{ext}}$ (e.g., $\phi_j^{\text{ext}} = \pi$ within a given connected interval, $j_0 < j < j_1$, and $\phi_j^{\text{ext}} = 0$ everywhere else, which gives the black and white hole horizons, see sections below), one simply needs to invert the square matrix $\alpha$. Generally, we note that for our purposes, the precision requirements are not so stringent: as long as deviations from the target phase profile are small (with respect to $\pi$),

the inductances in the array are still inverted, resulting in the desired apparent event horizon.

The computation of matrix $\alpha$ is in general a complicated numerical task. We can however provide a rough estimate by assuming the junctions to be arranged in a straight line, and applying the Biot-Savart law (similar in spirit to Ref. [117]). To this end, we assume that each current-carrying wire $j$ is described by a small current element of length $\Delta L$. For small current loops with radius $R$, we can simply set $\Delta L \sim R$ [118]. Assuming Coulomb gauge (equivalent to the irrotational gauge, up to screening of induced charges on the superconducting surfaces [69], which we here neglect for simplicity) we have to solve $\nabla^2 \mathbf{A} = -\mu_0 \mathbf{j}$ with the current density $\mathbf{j}$ for a small current carrying element. Assuming an infinitesimal, delta-like current element $\mathbf{j} \sim \delta(\mathbf{x})\mathbf{x}$, and plugging the resulting $\mathbf{A}$ into Eq. (22), we find

$$\alpha_{j,j'} \approx \frac{1}{I_0} \frac{R}{\sqrt{\Delta x^2 (j - j')^2 + \Delta l^2}}, \qquad (23)$$

where $\Delta x$ is the separation between Josephson junctions, and $\Delta l$ is the separation between current loop $j$ and junction $j$. The characteristic current $I_0$ is given as $I_0^{-1} = \mu_0 \mathcal{L}/(2\Phi_0)$, where $\mathcal{L}$ is the length of path $\mathcal{L}_j$ in Eq. (22). It represents the size of the fine structure containing the Josephson junction (Niemayer-Dolan bridge). For a typical size of $\mathcal{L} \sim 10\mu m$, we find $I_0 \sim 10\mu A$ (which is at the upper bound of the current that can usually pass through flux lines). Note that this is a crude upper bound for $I_0$. Very likely, $I_0$ is reduced significantly due to screening of the electric field which has here been neglected. We therefore expect that a flip of the phase by $\sim \pi$ is feasible. The other central figure of merit is the speed with which the phase flip can be performed. Defining the ramping time from zero current to target current for a single flux line as $\Delta t_I$, the phase flip is guaranteed to be nonadiabatic for $\sqrt{E_C E_J}\Delta t_I < 1$. This requires either the use of sub-GHz qubit designs, similar in spirit to a recent proposal in Ref. [72], or ultrafast ($\sim$ 100 gigasamples per second) arbitrary waveform generators. The latter may cause problems with usual Al-based architecture, but improvements were very recently demonstrated in connection with Nb-based circuits [119].

For the matrix given in Eq. (23), we can explicitly compute a required current profile that needs to be applied in order to swap the phases by $\pm\pi$ within a connected part of the 1D chain. Blue dots in Fig. 4 show that if $\Delta l \ll \Delta x$, $\alpha$ is almost perfectly diagonal, such that there is a one-to-one correspondence between the desired target phase profile and the required current profile that has to be applied to the individual loops. Note however that the onset of non-local effects may play a role even for (moderately) small $\Delta l/\Delta x$ (orange and brown dots in Fig. 4), where we need to apply a slightly nonlocal current profile to have a sharp, local switch from positive to negative inductances within the 1D chain. To conclude this section, we point out similarities and differences to previous proposals creating analog event

horizons and Hawking radiation in solid state systems. Specifically, the analog Hawking radiation previously predicted in Josephson junction arrays [32, 33] relies on the creation of solitons, where within the finite extension of the soliton (over many lattice sites), the inductance can likewise be regarded as effectively negative within the soliton profile. The procedure we produce on the other hand involves creating $\pm\pi$ shifts on a connected series of junctions, which in essence corresponds to a tightly packed generation of "half"-solitons with the width of a single lattice site. Our system thus seems arguably much less stable. But first, as indicated already, the nonreciprocity provided by the gyrators allows for a stabilization of the system even with inverted inductors, due to an extra (positive) inductive contribution. Second, the extra tilting mechanism provided by the gyrators allows for the explicit realization of arbitrary (on-demand) spacetime geometries, and in addition, the explicit creation of wormholes with (in principle) distinguishable black and white hole horizons. As for the evolution of the unstable system beyond transient times, we provide some thoughts in Sec. VI. Another important difference is that we here have in principle perfect control on the *position* of the horizon, contrary to the solitons studied in Ref. [32, 33], which are autonomously moving with respect to the lab frame. Moreover, we can create an event horizon with the precision of a single lattice site. This is in contrast to all other analog Hawking radiation proposals (cold atoms, circuits, tilted Weyl semimetals), where the analog metric always changes over a finite width (a healing length). This is an interesting caveat specifically regarding the existence of a well-defined Hawking temperature, or more generally, how to make sense of horizons with diverging curvature, as already indicated in Sec. II.

## V. ANALOG HORIZONS VIA FLUX QUENCH

Once the optimization problem of the current profile has been solved, the sign of the Josephson energy can be flipped for a region of the circuits described by the Hamiltonians (14) and (15), resulting in two horizons in the circuits (Fig. 1c). In the presence of horizons, the energy spectrum is in general complex for both nearest and next-to-nearest neighbour implementations.

The presence of tilted dispersion relations and horizons can be probed by means of injecting wave packets within the chains, and probe their time of flight. For example, within an overtilted region, wave packets can only move in one direction. At a white hole horizon, an incoming wave packet comes to a stand still. Such features represent a first experimentally available signature of the nontrivial spacetime geometry. Note that for the time evolution of the wave packets, the distinction between nearest and next-to-nearest neighbour coupling is not that important, under the condition that the experimenter is able to prepare wave packets close to $k \approx 0$ or $k \approx \pi$. These two types of excitations clearly distinguish between black and white hole horizon as they probe only a local part of $k$

space. In addition, aspects of the wormhole stability (or instability) are not immediately visible in the wave packet amplitude (until the moment when the evaporation starts changing the spacetime geometry itself, see also Sec. VI below).

We therefore present in addition a more sophisticated measurement which is able to probe radiation due to instabilities in a more direct fashion. Namely, this section examines how to characterize the system via two point charge and phase correlation functions. As foreshadowed above, while for a stable system, quantum fluctuations remain bounded over time, they start to diverge in the presence of complex eigenvalues. In the quadratic approximation, see Eq. (16), it is convenient to express the charge and phase operators for the circuits (14) and (15), in terms of bosonic operators

$$N_j = \frac{i}{\sqrt{2}} \left( a_j - a_j^\dagger \right), \tag{24}$$

$$\phi_j = \frac{1}{\sqrt{2}} \left( a_j + a_j^\dagger \right), \tag{25}$$

that obey the following commutation relations $[a_j, a_{j'}^\dagger] = \delta_{jj'}$, $[a_j, a_{j'}] = 0$. Now, calculating the two point correlation functions for charge and phase operators turns into calculating the correlation functions of the form $\langle a_j(t) a_{j'}^\dagger(t') \rangle$, $\langle a_j^\dagger(t) a_{j'}^\dagger(t') \rangle$, $\langle a_j(t) a_{j'}(t') \rangle$, and $\langle a_j^\dagger(t) a_{j'}(t') \rangle$.

We are going to calculate these correlations using two methods: 1) direct diagonalization and 2) an extension of Klich's determinant formula [120]. The reason for the multi-pronged approach will be explained in detail below. To summarize, 1) is in principle very straightforward to implement and program, and is more versatile, as it would allow computing all types of observables, not only correlations. But diagonalization is not applicable in the presence of exceptional points, which is where method 2) comes into play, since it only requires exponentiation of the matrix with exceptional points.

As previously stated, once the system has been quenched, its state is in a highly excited state, were the new ground state is removed from the theory (a well-defined procedure for times sufficiently close to the quench). In order to anchor the theory to a well-defined ground state (necessary to compute observables in a well-defined way), we choose to perform computations with respect to the ground state of the unquenched system. Since the diagonalization of the quenched Hamiltonian is being performed with respect to a state that is in general not its eigenstate,

we will need a generalised version of bosonic Bogoliubov transformations. These transformations are discussed in Appendix B.

Implementing the Bogoliubov transformations on the quenched Hamiltonian, mentioned in the previous paragraph, requires diagonalizing a non-Hermitian matrix. This can pose a problem if the matrix has defective eigenvalues (exceptional points). We therefore suggest another method of calculating the correlations after the quench. Consider the following way to write a two point correlation

$$\left\langle a_i^\dagger(t) a_j(t) \right\rangle = \partial_\chi \left\langle e^{\chi a_i^\dagger(t) a_j(t)} \right\rangle_{\chi=0}$$
$$= \lim_{\beta \to \infty} \frac{\text{Tr}\left( e^{i\mathcal{H}_> t} e^{\chi a_i^\dagger a_j} e^{-i\mathcal{H}_> t} e^{-\beta \mathcal{H}_<} \right)}{\text{Tr}\left( e^{-\beta \mathcal{H}_<} \right)}, \tag{26}$$

where $\mathcal{H}_<$ and $\mathcal{H}_>$ are the unquenched and quenched Hamiltonians respectively. The many body traces in the above expression can be calculated by an extension of Klich's trace-determinant formula. The original formula was introduced to calculate the trace of many body operators by replacing it by a determinant of the corresponding first quantized operator [120]. The extension of this formula is described in appendices C and D. This approach bypasses the problem posed by defective eigenvalues but it is also considerably slower than diagonalization for numerics, hence limiting the size of the systems we can work with. Another important remark: in the derivation (Appendix C) we have to assume the existence of a ground state for a non-Hermitian operator, which is not guaranteed. For some context about diagonalizing quadratic bosonic non-Hermitian operators we also refer the reader to Ref. [121], where the concept of third quantization [122] was extended to the bosonic Linblad equation. In Ref. [121], the existence of a ground state (or more appropriately, a non-equilibrium steady state) is always guaranteed due to the form of the Lindblad equation. Crucially, we do not have this constraint.

The system we will study here has fifty sites ($J = 50$) with the wormhole located between $j_0 = 15$ and $j_1 = 35$. For convenience, we impose periodic boundaries in position space (i.e., the 1D chain is closed into a loop). The parameters of the system are $|E_L'| = |E_L| = 3.42 E_C$, $G = 2.4$ and $M = 3 * 10^{-4} E_C$. As already pointed out, in each of the following figures of correlation functions the result could be obtained by both diagonalization and the generalized Klich determinant. The time steps in the following plots are $\delta t = 0.031 E_C^{-1}$ for the circuit with nearest neighbor coupling and $\delta t = 0.31 E_C^{-1}$ for the next-to-nearest neighbor coupling.

First, let us examine the plots of correlation functions for the circuit with only nearest neighbor coupling, Fig. 5. We see that both phase and charge quantum fluctuations

diverge after the quench. The system has two boundaries (red dotted lines) between the wormhole and the normal regions, that act as the horizons (located at $j_0$ and $j_1$).

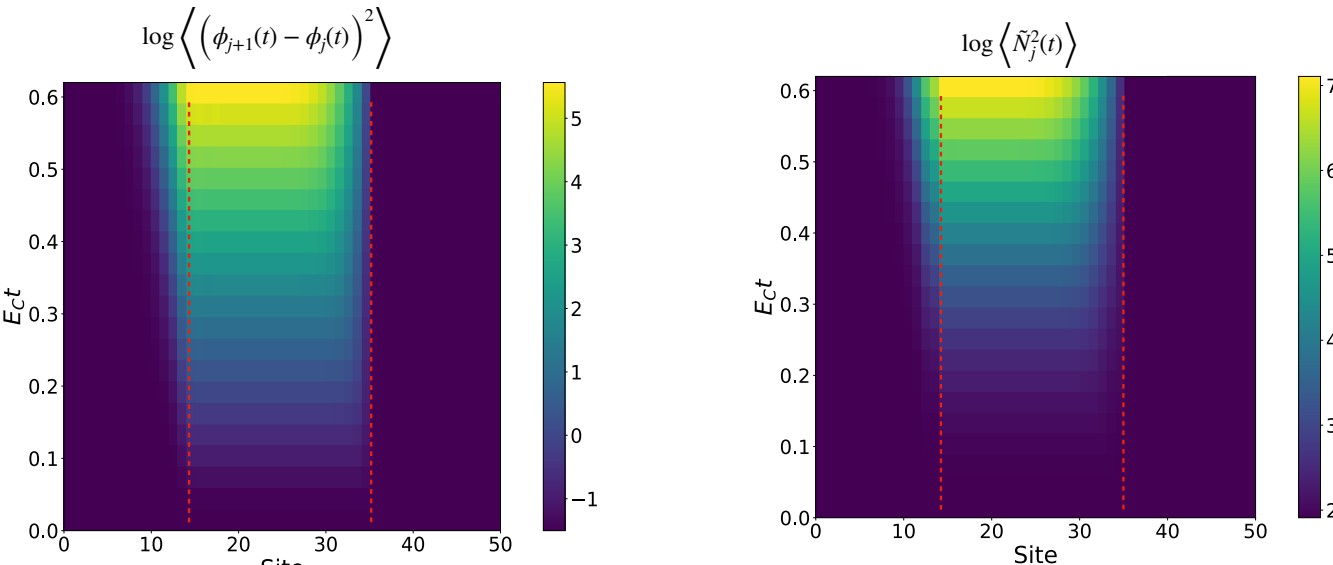

FIG. 5. System with Josephson junctions connecting the nearest neighbors, (left) time evolution of quantum fluctuations of phase difference, (right) time evolution of quantum fluctuations of conjugate charge. Red dotted lines denote the position of the apparent horizons, i.e., boundaries between the wormhole and normal regions. Here, the fluctuations visibly distinguish between a pure black (pure white) hole horizon, where quantum fluctuations accumulate (or not). More over, the interior between the two horizons is here unstable, such that quantum fluctuations diverge immediately within the entire wormhole region.

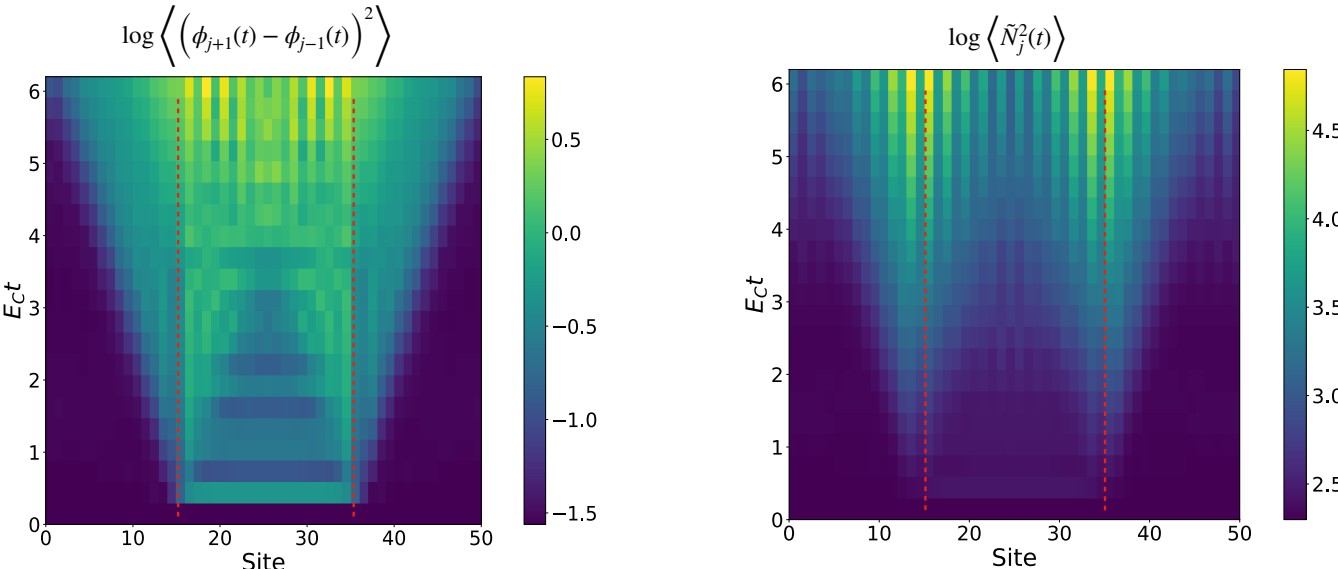

FIG. 6. System with Josephson junctions connecting the next-nearest neighbors, (left) time evolution of quantum fluctuations of phase difference, (right) time evolution of quantum fluctuations of conjugate charge. Red dotted lines denote the position of the horizons. Here, (and contrary to Fig. 5) both horizons have black as well as white hole character, such that the two horizons are not easily distinguised when considering the spatial dependence of the fluctuations. On the other hand, the wormhole interior is here (at least initially) stable, and quantum fluctuations grow only from the horizons outwards.

If we initialize low energy excitations (for this circuit this means excitations near $k = 0$) in the region with overtilted dispersion, they move from the left horizon to the right horizon, hence we label them as the black and the white hole horizon respectively. The black hole horizon radiates away with time, as can be inferred by the accumulation of quantum fluctuations near it with time (which indicates presence of radiation), while the white hole horizon does not radiate. This explicitly confirms the previously discussed fact that for the nearest neighbour system, black and white hole horizons are distinguishable. The wormhole region also starts radiating due to the presence of the exceptional point (as likewise already discussed in Sec. III). We observe that the decay in the

wormhole interior is much faster than the one induced by the presence of the horizons, which leads to the correlation functions growing much more rapidly *inside* the wormhole, before the black hole horizon show any appreciable decay. Overall, the above thus illustrates the possibility to create lattice simulations of wormholes with distinguishable black and white hole horizons, at the cost of a strong instability of the wormhole interior.

Now, let us focus on the circuit where Josephson junctions connect next nearest neighbors. Low-energy excitations in this circuit can be either near $k = 0$ or $k = \pm\pi$. For excitations (inside the wormhole region) near $k = 0$, similar to the previous circuit, the left boundary ($j_0$) acts as the black hole horizon and the right one ($j_1$) as the white hole. In contrast, for excitations near $k = \pm\pi$ the left boundary acts as the white hole horizon and right boundary as the black hole. This symmetry is also reflected in the correlation plots in Fig. 6, where we can observe both horizons radiating away identically. Also, the only instability in this circuit is due to the horizons, leading to a much slower collapse.

## VI.  PERSPECTIVE ON WORMHOLE EVAPORATION OVER LONG TIMES

As shown in the previous section, charge and phase fluctuations accumulate very rapidly due to the fundamental instability inherent to the wormhole horizons or interior (in case of nearest neighbour inductive coupling). In particular, the more the system accumulates charge and phase fluctuations over time, the more the quadratic approximation of the Josephson energy becomes inaccurate, such that predicting the system dynamics for long times becomes a much harder task (as the system can no longer be approximated by non-interacting bosons). This is well beyond the scope of the present work. We nonetheless find it illustrative to speculate on a qualitative level about the long term fate of the wormhole – especially as it allows us to distinguish between intrinsic (spontaneous) collapse, and dissipative relaxation. In addition, some of the resulting concepts provide an interesting segue to the subsequent idea: quantum superpositions of the spacetime geometry.

To this end, let us take into account interactions with an environment. For the sake of simplicity and concreteness, we consider the phase difference $\phi_j - \phi_{j-1} + \phi_{\text{ext},j}$ across a single junction. At the beginning of the wormhole quench, this phase difference is either localized around 0 (minimum of the cosine, if the junction is positioned outside the wormhole) or around $\pm\pi$ (maximum of the cosine, inside the wormhole). In the presence of the instability, the quantum fluctuations of this phase difference blow up over time, as shown in the previous section. Now suppose that the environment entangles (either weakly or strongly) with the current across this junction. This means that the phase difference gets spontaneously projected onto a more localized state. Notably, this type of environment-induced process thus *diminishes* quantum fluctuations. We note that the same picture holds in perfect analogy for the accumulation of charge noise quantum fluctuations, respectively, the reduction thereof by a dissipative process extracting information about the charge.

To proceed, note that the location of the wave function now is no longer at 0 or $\pi$, but can be (with a finite probability) at a certain distance from the minimum or maximum. With the new location updated, we again quadratically expand the cosine around the new peak position of the phase difference. Importantly, due to the cosine behaviour of the junction the resulting effective inductance is now different (the curvature of the cosine obviously changes as a function of the position at which it is calculated). Thus, as time progresses, and the phase difference starts to classically diffuse after repeated environment-induced measurements, the effective inductance at a given junction likewise fluctuates classically. At this point, the impact of such dissipative processes could likewise be detectable in time-of-flight measurements of wave packets within the array. At any rate, the most likely long-term outcome is that all phase differences relax to the ground state, equivalent to the complete evaporation of the analog wormhole. We note that for a system on a ring (with periodic boundaries), or a system with a finite mass term (Cooper-pair leakage to ground), the emergence of soliton states might be a possibility.

Let us repeat that even though the environment will undoubtedly have a big impact on the system dynamics (especially beyond the immediate transient time scale), the experimental signatures could not be more different. Analog Hawking radiation leads to an increase of *quantum* fluctuations, whereas the environment induces *classical* fluctuations. Thus, the observation of the former can distinguish between how the system intrinsically reacts to the creation of an instability, as compared to the impact of external perturbations.

To conclude this section, we consider the very hypothetical case of negligible coupling to the environment, as it reveals an interesting additional idea which we pursue in the remainder of this work. If we assume that the build-up of quantum fluctuations of the phase could progress unimpeded over longer times, the nonlinearity of the Josephson energies results not in classical fluctuations of the effective inductance (as in the previous paragraph), but in a system that has to be interpreted as being in a *quantum* superposition of different effective inductances – a form of quantum superposition of the spacetime geometry itself. However, within the above considered setting, this effect is likely not to survive for very long due to environment-induced decoherence of the quantum fluctuations, and even for a highly protected system, it would be hard to analyse the effect in an unambiguous way. In what follows now, we provide a modification of this idea, not relying on instabilities (and thus not requiring any ultra-fast quenches), but instead on more general notions of Josephson junctions with multiple minimas.

This endeavour will on the one hand introduce the notion of a quantum inductance, and with it, a likely more stable version of a quantum superposition of the analog spacetime.

## VII. BISTABLE JOSEPHSON EFFECT AND QUANTUM SPACETIME GEOMETRY

As announced, in this section we propose a possible way to create and observe superpositions of two different spacetime geometries using quantum circuits. As already hinted at in the introduction, the idea that spacetime geometries can exist as a superposition within actual gravity is exciting, but at the same time highly contentious [54].

Note, moreover, that the currently most debated possibility is the observation of entanglement created between two quantum systems via the gravitational force, hence proving the quantum nature of gravitation [50, 51]. Since gravity, in part, is a theory of the structure of spacetime, it being quantum in nature may imply that spacetime can exist in quantum superposition. The notion that we introduce now is insofar related, as it allows for the formulation of a quantum superposition of analog spacetime geometry, by exploiting internal degrees of freedom of general non-linear inductor elements.

The central idea can be formulated as follows. Previously, the local signal speed in a circuit composed of Josephson junctions, is uniquely defined by the curvature of the minimum of the Josephson potential. However, if we use a potential with more than one minimum, where – crucially – each minimum has a different curvature, they will have different signal speed. Since these two minima can be considered a quantum double well, it is possible to prepare a state where one occupies a superposition of the two minima, ultimately allowing us to create a quantum superposition of signals travelling simultaneously at multiple different speeds.

Generalized Josephson junctions with a more complicated energy-phase relationship, with, e.g., multiple minima (so-called multistable Josephson junctions) have been studied for some time by now. The simplest one is a junction whose dominant process consists of the tunnelling of *pairs* of Cooper-pairs, described by an energy with the phase dependence $\sim \cos(2\phi)$ [100, 101]. Their design as proposed in Ref. [101] is shown in Fig. 8(a). This design can also be generalized to get an element with the phase dependence $\sim \cos(3\phi)$ (Fig. 8(b)) [123]. We now imagine a parallel shunt of a regular Josephson junction with these generalized junctions to realize a potential of the form (Fig. 7)

$$V \sim -\cos 2\phi + \delta \cos \phi - \delta \cos 3\phi, \qquad (27)$$

with $|\delta| < 1/2$. This function has two minima, whose curvatures are $\sim 4(1 \pm 2\delta)$. We note that while the parallel shunt of the above three different junction types

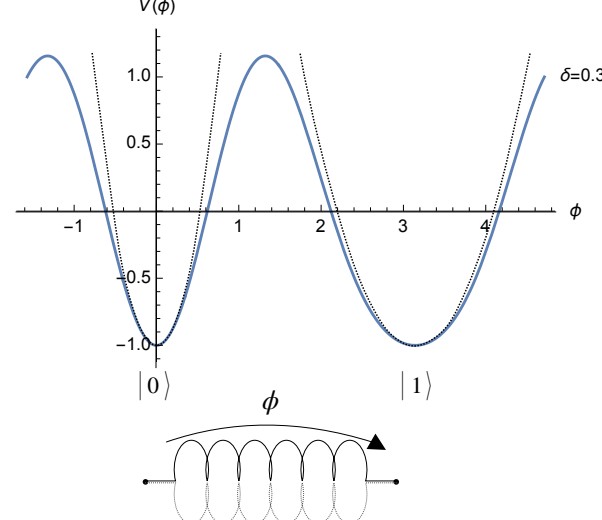

FIG. 7. Realization of a quantum inductor circuit element. We need an energy-phase relationship, Eq.(27), with two minima (multi-stable junction) of unequal curvatures, resulting in two different inductances. We label the two minima as the basis states of a qubit $|0\rangle$ and $|1\rangle$. When preparing the multistable junction in a superposition of the two minimas, the resulting inductance is likewise in a quantum superposition of two values. The inset figure, below the graph, shows the circuit symbol for a quantum inductor.

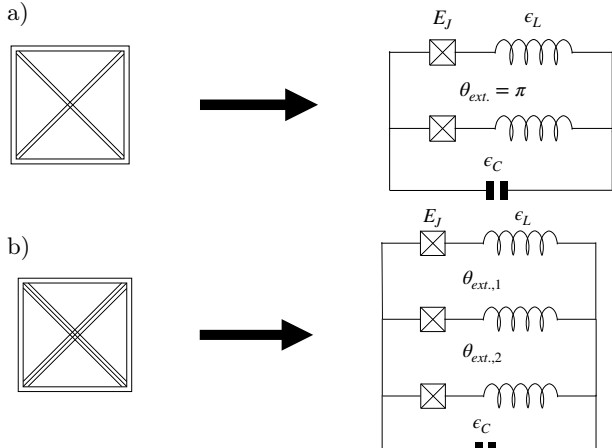

FIG. 8. Realization of a junction with $\cos(2\phi)$ energy profile (transporting pairs of Cooper pairs, see also Ref. [101]) in panel a). A $\cos(3\phi)$ element is realized according to the layout in b). In the circuit diagrams above for the generalized Josephson junctions, the Josephson junctions have the same parameters and so do the inductors. The shunt capacitance ($\epsilon_C$) is the dominant energy scale.

is very instructive to demonstrate the principle, the actual experimental fabrication of a potential of the form of Eq. (27) may be simplified. For instance, it is expected [123] that the setup realizing the term $\sim \cos(3\phi)$ [Fig. 8(b)] has enough free parameters to implement the potential without the additional inclusion of a $\cos(\phi)$ and a $\cos(2\phi)$ junction.

At any rate, the above potential has the sought-after minima with different curvatures. This circuit element is therefore at low energies described as a type of quantum inductor, which can exhibit superpositions of different inductance values. Using such an element to connect nodes $j$ and $j-1$, this element can be described by the energy

$$(E_L + \delta E_L \sigma_j^z)(\phi_j - \phi_{j-1})^2 \, , \qquad (28)$$

where the relative difference of the inductances is given by $\delta E_L/E_L = 2\delta$ [with $\delta$ as it appears in Eq. (27)], and the pseudo-spin $\sigma^z$ represents the two distinct quantum well states of the inductor, $|0_j\rangle$ and $|1_j\rangle$, having eigenvalue $-1$ and $+1$, respectively. We note that this description is of course a crude approximation. In particular, finite capacitances will likely induce transitions $\sim \sigma_j^x$. While we for now discard this possibility for simplicity, we will outline in the outlook, that the inclusion of coherent spin-flip processes (or even nonlocal spin-spin interactions) by no means pose a danger to the proposal outlined here, but to the contrary, render it even more interesting (as they may give rise to a rudimentary simulation of an analog quantum spacetime with an intrinsic coherent dynamics of its own).

The full proposal to realize and measure superpositions of spacetime geometries now works as follows. Consider the circuit in Fig. 9 consisting of three regions, two regions where the superconducting nodes are connected by Josephson junctions and a third one sandwiched between the the other two where superconducting nodes are connected by the newly introduced quantum inductors. This system is represented by the following Hamiltonian

$$\tilde{\mathcal{H}} = \sum_j E_C N_j^2 + \sum_{j=j'}^{j''} \left( E_L + \delta E_L \sigma_j^z \right) (\phi_{j+1} - \phi_j)^2$$
$$+ \sum_{j \notin [j', j'']} E_L' (\phi_{j+1} - \phi_j)^2 \, . \qquad (29)$$

For our proposal the circuit needs to be prepared in the maximally entangled state $\left( |\tilde{0}\rangle + |\tilde{1}\rangle \right)/\sqrt{2}$, where

$$|\tilde{0}\rangle = |0_{j'}0_{j'+1}...0_{j''-1}0_{j''}\rangle \equiv \begin{pmatrix} 1 \\ 0 \end{pmatrix}$$

$$|\tilde{1}\rangle = |1_{j'}1_{j'+1}...1_{j''-1}1_{j''}\rangle \equiv \begin{pmatrix} 0 \\ 1 \end{pmatrix}.$$

To measure this entanglement we will exploit the fact that the two states of the asymmetric potential correspond to different signal speeds, present in the sandwiched region. Let us now initialize a wave packet in the trivial region to the left. Assuming for concreteness a signal with large amplitude in $\phi$, we can simply describe it with a wave function $\phi(x-ut)$ (using a continuous space representation $x = \Delta x j$, valid if the wave packet is large with respect to the lattice size $\Delta x$), centered around a position on the left, and moving towards the region with the quantum inductors. The propagation speed here is $u = \Delta x \sqrt{E_C E_L'}$. Within this region, the signal will now propagate with two different speeds. If the quantum inductor region is large enough with respect to the speed difference and the wave packet size, we will create a fully entangled state. The transition from the incoming wave packed to the left of the quantum inductor region, and the outgoing wave packet to the right of the quantum inductor region can be described as

$$\frac{1}{\sqrt{2}} \begin{pmatrix} \phi(x-ut) \\ \phi(x-ut) \end{pmatrix} \rightarrow \frac{1}{\sqrt{2}} \begin{pmatrix} \phi(x+x_0-ut) \\ \phi(x+x_1-ut) \end{pmatrix} . \qquad (30)$$

The positions $x_0$ and $x_1 > x_0$ (i.e., the wave packet $\phi(x+x_1-ut)$ is the further advanced one) express the different positions of the outgoing wave packets due to the different times of flight within the quantum inductor region. If the overlap vanishes, i.e. if $\phi(x+x_0-ut)\phi(x+x_1-ut) \approx 0$, the entanglement is maximal.

A correlated measurement of the wave packet (e.g., time of flight measurement) and the state of the circuit ($|\tilde{0}\rangle$ or $|\tilde{1}\rangle$) provides information about the presence of the entanglement. Note that in order to verify the quantum coherent nature of the analog spacetime, one can easily formulate a more sophisticated interference setup. To this end, after the wave packets have traversed the quantum inductor region, we perform a Hadamard gate on the qubit states, and add a hard wall on the far right end, such that the further advanced wave packet can reflect. This sequence yields the state

$$\frac{1}{\sqrt{2}} \begin{pmatrix} \phi(x+x_0-ut) \\ \phi(x+x_1-ut) \end{pmatrix} \xrightarrow{\text{Hadamard}} \frac{1}{2} \begin{pmatrix} \phi(x+x_0-ut) + \phi(x+x_1-ut) \\ \phi(x+x_0-ut) - \phi(x+x_1-ut) \end{pmatrix} \qquad (31)$$

$$\xrightarrow{\text{hard wall}} \frac{1}{2} \begin{pmatrix} \phi(x+x_0-ut) + \phi(x+\tilde{x}_1+ut) \\ \phi(x+x_0-ut) - \phi(x+\tilde{x}_1+ut) \end{pmatrix} , \qquad (32)$$

where in the last line, the updated shift $\tilde{x}_1$ takes into account the distance travelled up to and after collision

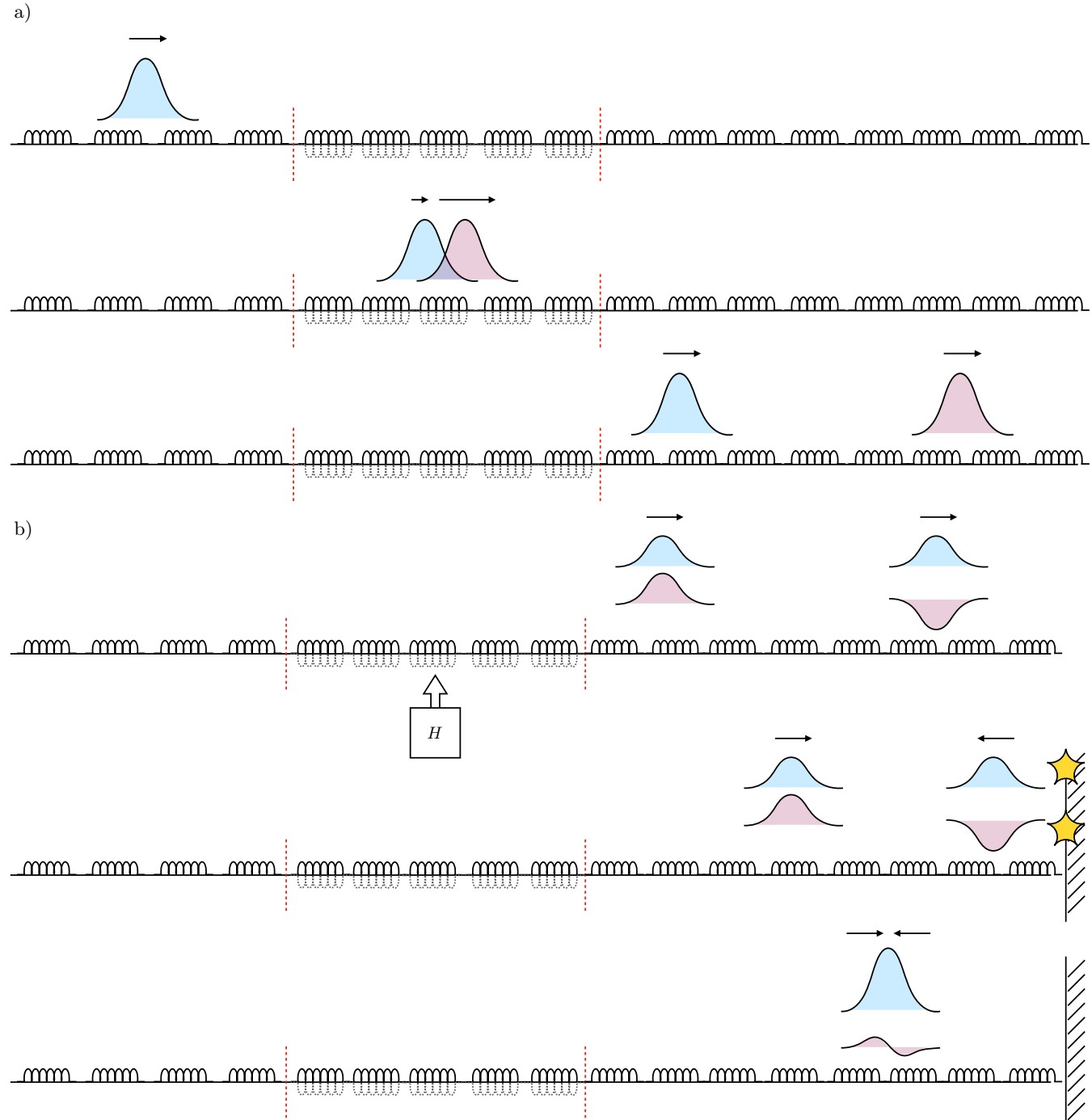

FIG. 9. Illustration of quantum superposition of spacetime geometry and its detection with travelling signals in snapshots. a) Initializing a signal on the left hand side (with ordinary classical flat spacetime), the signal starts to separate in the quantum spacetime region due to different time of flight, and ultimately exits in the normal region on the right in a fully entangled state. b) After pplying a Hadamard gate on the maximally entangled state in the quantum spacetime region, and letting the signals reflect at a hard wall (yellow stars indicate the reflection event) the signals interfere either constructively or descructively.

with the hard wall. As can be seen on the last line of the above equation, the further advanced package now travels backwards ($+ut$ instead of $-ut$), and is thus bound to interfere with the delayed wave packet (which still travels forward, $-ut$). In this setup, interference is constructive for the $|\tilde{0}\rangle$ branch, and destructive for the $|\tilde{1}\rangle$ branch. Hence, by projective measurement of the signal either directly after passing the quantum inductor region, or measuring after the succession of Hadamard gate and reflection, the quantum nature of the spacetime geometry can be unequivocally probed.

## VIII. CONCLUSIONS AND OUTLOOK

In this paper we have proposed using quantum circuits to implement two aspects of quantum gravity: 1) analog horizons and 2) quantum superpositions of spacetime geometries. Even though the former idea is a well explored concept in solid state systems in general, we here explicitly demonstrate the capacity of superconducting circuits to emulate arbitrary spacetime configurations. By identifying a minimal fundamental set of necessary circuit elements, we further unravel a number of surprising findings pertinent to lattice systems, which allow in particular to create horizons with trans-Planckian curvature, allowing us to disentangle the effect of curvature of background metric from the effect of a horizon – and in doing so, exploring the extreme regime of horizons with diverging curvatures, which as far as we know has not been done in any existing proposal. To implement the analog horizons in a quantum circuit, we proposed a general way of creating negative inductances with Josephson junctions transiently driven by nearby current loops. We further uncovered two subtly distinct ways to engineer a region with overtilted dispersion relation, either with inductive coupling of nearest or next-to-nearest neighbour nodes. While for the latter, horizons have a combination of black and white hole character (as is the most usual case in lattices), the former [Eq.(14)] hosts two distinct kinds of horizons, one that acts purely as a black hole horizon and the other as a white hole horizon. For this second type of circuit, it is not only the horizons, but also the region between the horizons (wormhole interior) that is unstable due to the presence of an exceptional point. This instability also contributes to the observed radiation (in addition to the evaporation of the horizons) as can be confirmed by tracking the quantum fluctuations of phase difference and conjugate charge with time (Fig. 5). In addition to Bogoliubov diagonalization, we used an extension of Klich's trace-determinant formula to obtain the correlation functions numerically. The proof of this extension remains elusive for a general operator, but in the parameter regime where the generalised Bogoliubov transformation is applicable, the results of both numerical methods agree.

Our second proposal concerns the controversial idea of superpositions of spacetime geometries. This idea was shown to emerge as a nontrivial extension to the above negative inductance realization. The emulation of such geometries relies on engineering junction elements with an energy-phase relationship that exhibits an asymmetric double well potential (Eq.(27)), so-called multi-stable Josephson junctions. The different curvatures of the two minima of the potential give us the two inductances that act as the basis states for the quantum inductance. We proposed a setup where a classical signal traverses the quantum spacetime region, resulting in a superposition of two different signal speeds. The superposition can be measured by making correlated measurements of the quantum inductance and the signal moving through the circuit, or by means of a delayed choice experiment.

In this paper, in addition to standard circuit elements, we have also included elements that are still being developed experimentally (such as quantum gyrators), or introduced new effective circuit elements (based on regular or multistable Josephson junctions), which, even though experimentally tested in existing works, are here proposed to be operated in previously unexplored regimes. We here therefore point at feasible small scale experimental tests that can act as proofs of principle for the basic physical principles behind the considered physics. For instance, our proposal of using current loops to invert the effective inductance of a Josephson junction can be tested on a single transmon, where the qubit frequency becomes imaginary $\sqrt{E_J E_C} \rightarrow \sqrt{-E_J E_C}$ for transient times. The resulting accumulation of phase or charge quantum fluctuations (or limits thereof due to coupling to the environment) can thus be tested in an "imaginary" qubit device. Such a study might be also of some fundamental interest, when extending the description of the junction beyond the quadratic approximation, where the expansion around the energy minimum or the energy maximum can be regarded as real and imaginary twins of nonlinear dynamics – ready to be explored in future works. For the case of quantum inductors, one can likewise consider a simple circuit with either one or a few quantum inductors in series, where the low energy excitations induced in the two minima will have different oscillation frequency – allowing for the engineering of (nearly) harmonic oscillators in a quantum superposition of two resonance frequencies.

Finally, we comment on extensions towards a more fully-fletched simulation of analog quantum spacetimes. Namely, in our treatment in the main text, the states of the quantum inductors had, as of now, no intrinsic dynamics. While this is the simplest possilbe treatment from a theoretical stand point, in experiment it would require a tuning to very extreme ratios of small charging energy compared to the junction energy scale. For more moderate ratios, it cannot be excluded that the basis states of the quantum inductors receive an intrinsic dynamics, for example a tunnel coupling between the two minima, $\sigma_j^x$, or non-local couplings such as $\sigma_j^\alpha \sigma_{j'}^\beta$ $(\alpha, \beta = x, y, z)$, which have to be added in the Hamiltonian in Eq.(29). In this regard, note that this specific quantum spacetime proposal is in line with currently existing plans to engineer quantum computing hardware by means of flux qubits based on multi-stable Josephson junctions [100–102]. Crucially, however, while the above described type of crosstalk is obviously detrimental for the performance of the hardware when used for standard quantum information applications, for the here considered analog quantum gravity simulation, such imperfections are by no means a disadvantage – to the contrary, they may open up a new road to explore analog quantum gravity where the spacetime metric has its quantum coherent dynamics.

## IX. ACKNOWLEDGEMENTS

We acknowledge interesting and fruitful discussions with Zaki Leghtas, Roudy Hanna, Philippe Campagne-Ibarcq and David DiVincenzo. This work has been funded by the German Federal Ministry of Education and Research within the funding program Photonic Research Germany under the contract number 13N14891.

### Appendix A: Josephson junction array with gyrators

Here we solve a JJ array with gyrators

$$\mathcal{H} = \sum_{j=1}^{J} \left[ N_j + G(\phi_{j+1} - \phi_{j-1}) \right]^2 + E_L \left( \phi_{j+1} - \phi_j \right)^2 + M\phi_j^2 , \tag{A1}$$

and periodic boundary conditions, where we have introduced a mass term to regularize the zero mode. Define the fourier transform as follows

$$N_k = \frac{1}{\sqrt{J}} \sum_{j=1}^{J} e^{ik_m j} N_j ,$$

$$\phi_k = \frac{1}{\sqrt{J}} \sum_{j=1}^{J} e^{-ik_m j} \phi_j ,$$

$$k_m = \frac{2\pi m}{J} , \tag{A2}$$

where $m \in \{-J/2 + 1, -J/2 + 2, ..., J/2\}$. The Hamiltonian can now be written as

$$\mathcal{H} = E_C \sum_{m=-\frac{J}{2}+1}^{\frac{J}{2}} \left[ N_m N_{-m} + i2G \sin\left( \frac{2\pi m}{J} \right) \{N_m, \phi_m\} + \beta_m \phi_m \phi_{-m} \right] , \tag{A3}$$

where

$$\beta_m = \frac{4E_C G^2 \sin^2\left( \frac{2\pi m}{J} \right) + 4E_L \sin^2\left( \frac{\pi m}{J} \right) + M}{E_C}.$$

Now, we write the Fourier transformed charge and phase operators in terms of bosonic annihilation and creation operators

$$N_m = \frac{i}{\sqrt{2}} \left( a_{-m} - a_m^\dagger \right) , \tag{A4}$$

$$\phi_m = \frac{1}{\sqrt{2}} \left( a_m + a_{-m}^\dagger \right) , \tag{A5}$$

that obey the following commutation relations $[a_m, a_{m'}^\dagger] = \delta_{mm'}$, $[a_m, a_{m'}] = 0$. Finally, the Hamiltonian becomes

$$\mathcal{H} = \frac{E_C}{2} \sum_{m=1}^{\frac{J}{2}-1} \left( a_m^\dagger \ a_{-m}^\dagger \ a_m \ a_{-m} \right) H_m \begin{pmatrix} a_m \\ a_{-m} \\ a_m^\dagger \\ a_{-m}^\dagger \end{pmatrix} + \frac{E_C}{2} \left( a_0^\dagger \ a_0 \right) h \begin{pmatrix} a_0 \\ a_0^\dagger \end{pmatrix} + \frac{E_C}{2} \left( a_{\frac{J}{2}}^\dagger \ a_{\frac{J}{2}} \right) h' \begin{pmatrix} a_{\frac{J}{2}} \\ a_{\frac{J}{2}}^\dagger \end{pmatrix} ,$$

$$\mathcal{H} = \sum_{m=1}^{\frac{J}{2}-1} \mathcal{H}_{1,m} + \mathcal{H}_2 + \mathcal{H}_3 , \tag{A6}$$

where

$$H_m = (\beta_m + 1) \, \mathbb{I}_4 + 4G \sin\left( \frac{2\pi m}{J} \right) \begin{pmatrix} \sigma_z & 0 \\ 0 & \sigma_z \end{pmatrix} + (\beta_m - 1) \begin{pmatrix} 0 & \sigma_x \\ \sigma_x & 0 \end{pmatrix} , \tag{A7}$$

$$h = (\beta_0 + 1) \, \mathbb{I}_2 + (\beta_0 - 1) \, \sigma_x , \tag{A8}$$

$$h' = \left( \beta_{\frac{J}{2}} + 1 \right) \mathbb{I}_2 + \left( \beta_{\frac{J}{2}} - 1 \right) \sigma_x. \tag{A9}$$

$\mathbb{I}_n$ is a $n$-dimensional identity matrix while $\sigma_z$ and $\sigma_x$ are Pauli matrices. We will focus on diagonalization of $\mathcal{H}_{1,m}$ in eq.(A6), the same formalism can be used for the other parts.

We will use the diagonalization of $\mathcal{H}_{1,m}$ to outline some general properties of a Bosonic Bogoliubov transformation. Our goal is to find new operators, $b_m, b_m^\dagger$ such that

$$\mathcal{H}_{1,m} = \frac{E_C}{2} \begin{pmatrix} b_m^\dagger & b_{-m}^\dagger & b_m & b_{-m} \end{pmatrix} \tau_z \Lambda_m \begin{pmatrix} b_m \\ b_{-m} \\ b_m^\dagger \\ b_{-m}^\dagger \end{pmatrix}, \tag{A10}$$

where $\Lambda_m$ is a diagonal matrix, and the new operators still satisfy the Bosonic commutation relations. This entails finding a matrix $Q_m$ such that

$$\Lambda_m = Q_m^{-1} \tau_z H_m Q_m, \tag{A11}$$

where

$$\tau_z = \begin{pmatrix} \mathbb{I}_2 & 0 \\ 0 & -\mathbb{I}_2 \end{pmatrix}.$$

Since the column vector and the row vector in eq.(A10) are related by Hermitian conjugation, this imposes the following condition

$$Q_m^{-1} = \tau_z Q_m^\dagger \tau_z. \tag{A12}$$

Moreover the column (and the row) in eq.(A10) has internal structure as well, the bottom half of the column contains Hermitian conjugates of the upper half, this results in the following conditions

$$Q_m^* = \tau_x Q_m \tau_x, \tag{A13}$$
$$\Lambda_m = -\tau_x \Lambda_m \tau_x, \tag{A14}$$

where

$$\tau_x = \begin{pmatrix} 0 & \mathbb{I}_2 \\ \mathbb{I}_2 & 0 \end{pmatrix}.$$

Finally, we can write

$$\mathcal{H}_{1,m} = \frac{E_C}{2} \begin{pmatrix} a_m^\dagger & a_{-m}^\dagger & a_m & a_{-m} \end{pmatrix} \tau_z Q_m \tau_z \tau_z \Lambda_m Q_m^{-1} \begin{pmatrix} a_m \\ a_{-m} \\ a_m^\dagger \\ a_{-m}^\dagger \end{pmatrix} \tag{A15}$$

$$= \frac{E_C}{2} \begin{pmatrix} b_m^\dagger & b_{-m}^\dagger & b_m & b_{-m} \end{pmatrix} \tau_z \Lambda_m \begin{pmatrix} b_m \\ b_{-m} \\ b_m^\dagger \\ b_{-m}^\dagger \end{pmatrix}. \tag{A16}$$

The above procedure of Bogoliubov transformation is only applicable if $\Lambda_m$ is a real matrix, for complex matrix we will develop the procedure in the next section.

The complete diagonalized Hamiltonian is

$$\mathcal{H} = \sum_{m=-\frac{J}{2}+1}^{\frac{J}{2}} \omega_m b_m^\dagger b_m, \tag{A17}$$

$$\omega_m = 2\sqrt{E_C^2 \beta_m + 4 E_C G \sin\left(\frac{2\pi m}{J}\right)}. \tag{A18}$$

We also investigate the arrays where the Josephson junction connects next nearest neighbors, the Hamiltonian for such a system will be

$$\mathcal{H}' = \sum_{j=1}^{J} \left[ N_j + G(\phi_{j+1} - \phi_{j-1}) \right]^2 + E_L \left( \phi_{j+1} - \phi_{j-1} \right)^2 + M \phi_j^2. \tag{A19}$$

This Hamiltonian can be diagonalized in a similar way to get

$$\mathcal{H}' = \sum_{m=-\frac{J}{2}+1}^{\frac{J}{2}} \omega_m b_m^\dagger b_m, \tag{A20}$$

$$\omega_m = 2\sqrt{E_C^2 \beta_m + 4 E_C G \sin\left(\frac{2\pi m}{J}\right)}, \tag{A21}$$

$$\beta_m = \frac{4\left(E_C G^2 + E_L\right)\sin^2\left(\frac{2\pi m}{J}\right) + M}{E_C}.$$

## Appendix B: Bogoliubov transformation for calculating correlations after quench

Consider the following Hamiltonian

$$\mathcal{H} = \sum_{j=1}^{J} \left[N_j + G(\phi_{j+1} - \phi_{j-1})\right]^2 + E_{L,j}\left(\phi_{j+1} - \phi_j\right)^2 + M\phi_j^2. \tag{B1}$$

The term $E_{L,j}$ indicates that the inductance can vary with site. Since there is no translational invariance, we cannot use Fourier transform, instead we write the phase and charge operators in terms of Bosonic annihilation and creation operators

$$N_j = \frac{i}{\sqrt{2}}\left(a_j - a_j^\dagger\right), \tag{B2}$$

$$\phi_j = \frac{1}{\sqrt{2}}\left(a_j + a_j^\dagger\right), \tag{B3}$$

that obey the following commutation relations $[a_j, a_{j'}^\dagger] = \delta_{jj'}$, $[a_j, a_{j'}] = 0$. This allows us to write the Hamiltonian as

$$\mathcal{H} = \mathbf{a}^\dagger H \mathbf{a}, \tag{B4}$$

where $\mathbf{a} = \begin{pmatrix} a_1 & \cdots & a_J & a_1^\dagger & \cdots & a_J^\dagger \end{pmatrix}^T$.

We begin with the unquenched Hamiltonian, where $E_{L,j} = E_L$, this is the same Hamiltonian that we solved in appendix A, this time in position space, we can still follow the same steps and diagonalize it without going to the Fourier space to get

$$\begin{aligned}
\mathcal{H}_< &= \mathbf{a}^\dagger H_< \mathbf{a} \\
&= \mathbf{a}^\dagger ZQZZ\Lambda_< Q^{-1}\mathbf{a} \\
&= \mathbf{b}^\dagger Z\Lambda_< \mathbf{b} \tag{B5} \\
&= \sum_{j=1}^{J}\left(\lambda_{<,j} b_j^\dagger b_j + \lambda_{<,j} b_j b_j^\dagger\right). \tag{B6}
\end{aligned}$$

Here

$$Z = \begin{pmatrix} \mathbb{I}_J & 0 \\ 0 & -\mathbb{I}_J \end{pmatrix},$$

and operators $\mathbf{b}^{(\dagger)}$ have the same commutation relations as $\mathbf{a}^{(\dagger)}$. The matrix $Q$ satisfies the conditions in eq.(A12) and (A13), while the diagonal matrix $\Lambda_<$ satisfies the eq.(A14). In position space the Hamiltonian does not break down into independent blocks of $4*4$ matrices, therefore the size of the matrices in the above equations have to be changed appropriately.

Now we quench the system such that

$$E_{L,j} = \begin{cases} -E_L & j_0 < j < j_1, \\ E_L & \text{everywhere else,} \end{cases} \tag{B7}$$

for some arbitrary $j_0$ and $j_1$. The quenched Hamiltonian can be written as

$$\mathcal{H}_> = \mathbf{a}^\dagger H_> \mathbf{a}$$
$$= \mathbf{a}^\dagger ZPZZ\Lambda_> P^{-1}\mathbf{a}. \tag{B8}$$

The diagonal matrix $\Lambda_>$ is not necessarily real, therefore the eq.(A12) and (A13) doesn't hold, but these conditions translate to conditions on the first quantized matrix as

$$H_> = H_>^\dagger, \tag{B9}$$
$$H_>^T = XH_>X, \tag{B10}$$
$$X = \begin{pmatrix} 0 & \mathbb{I}_J \\ -\mathbb{I}_J & 0 \end{pmatrix}.$$

which still hold true. From this we can get some new properties

$$P^{-1} = ZXP^TXZ, \tag{B11}$$
$$\Lambda_> = -X\Lambda_>X, \tag{B12}$$
$$\Lambda_> = M^{-1}\Lambda_>^* M, \tag{B13}$$

where $M = ZP^\dagger ZP$. One might notice that eq.(B11) is just the combination of eq.(A12) and (A13). The unquenched Hamiltonian can now be written as

$$\mathcal{H}_> = \mathbf{c}^\bullet Z_{2J}\Lambda_> \mathbf{c} \tag{B14}$$
$$= \sum_{j=1}^J \left( \lambda_{>,j} c_j^\bullet c_j + \lambda_{>,j} c_j c_j^\bullet \right), \tag{B15}$$

where

$$\mathbf{c}^\bullet = \begin{pmatrix} c_1^\bullet & \cdots & c_J^\bullet & c_1 & \cdots & c_J \end{pmatrix}$$
$$\mathbf{c} = \begin{pmatrix} c_1 & \cdots & c_J & c_1^\bullet & \cdots & c_J^\bullet \end{pmatrix}^T.$$

The new operators obey the following commutation relations $[c_j, c_{j'}^\bullet] = \delta_{jj'}$, $[c_j, c_{j'}] = 0$, $[c_j^\bullet, c_{j'}^\bullet] = 0$. These relations closely resemble Bosonic commutation relations, but it is important to note that the operators $c_j^\bullet$ and $c_j$ are not related by Hermitian conjugation.

Any two point correlation of charge and phase operators can be written as a linear combination of two point correlators of the $\mathbf{a}^{(\dagger)}$ operators, hence we focus on finding the following correlation matrix

$$F(t,t') \equiv \left\langle \mathbf{a}(t)\mathbf{a}^\dagger(t') \right\rangle, \tag{B16}$$

where the average is taken over the ground state of the equilibrium system. For a total number of $J$ sites $F$ is a $2J * 2J$ matrix with the substructure form

$$F(t,t') = \begin{pmatrix} A(t,t') & B(t,t') \\ C(t,t') & D(t,t') \end{pmatrix}, \tag{B17}$$

where $A, B, C$ and $D$ are block matrices of dimensions $J * J$ that contain the correlations of the form $\left\langle a_j(t)a_{j'}^\dagger(t') \right\rangle$, $\left\langle a_j^\dagger(t)a_{j'}^\dagger(t') \right\rangle$, $\left\langle a_j(t)a_{j'}(t') \right\rangle$ and $\left\langle a_j^\dagger(t)a_{j'}(t') \right\rangle$ respectively.

First let us calculate the correlation matrix for unquenched Hamiltonian

$$F_<(t,t') = Q\left\langle \mathbf{b}(t)\mathbf{b}^\dagger(t') \right\rangle ZQ^{-1}Z. \tag{B18}$$

In this step we have used the fact that the matrices $Q$ and $Z_{2J}$ commute with many-body operator $\mathcal{H}_<$. Since the ground state of unquenched Hamiltonian is defined as $b_i |0\rangle = 0 \ \forall i$, we get

$$F_<(t,t') = Qe^{-i2(t-t')\Lambda_<}\frac{\mathbb{I}_{2J} + Z}{2}Q^{-1}Z. \tag{B19}$$

Now we focus on the case where we quench the system. Namely, we prepare the system in a ground state of a different Hamiltonian $\mathcal{H}_<$ for times $t < 0$, and immediately switch the system to the Hamiltonian $\mathcal{H}_>$ at time $t = 0$ (and let it evolve for subsequent $t > 0$). Using the previous results we can finally write down the correlation matrix

$$
\begin{aligned}
F_>(t,t') &= \langle \mathbf{a}(t)\mathbf{a}^\dagger(t') \rangle \\
&= P \langle \mathbf{c}(t)\mathbf{c}^\bullet(t') \rangle Z P^{-1} Z \\
&= P e^{-i2\Lambda_> t} \langle \mathbf{c}\mathbf{c}^\bullet \rangle e^{i2\Lambda_> t'} Z P^{-1} Z \\
&= P e^{-i2\Lambda_> t} P^{-1} \langle \mathbf{a}\mathbf{a}^\dagger \rangle Z P e^{i2\Lambda_> t'} P^{-1} Z \\
&= P e^{-i2\Lambda_> t} P^{-1} Q \langle \mathbf{b}\mathbf{b}^\dagger \rangle Z Q^{-1} P e^{i2\Lambda_> t'} P^{-1} Z \\
&= P e^{-i2\Lambda_> t} P^{-1} Q \frac{\mathbb{I}_{2J} + Z}{2} Q^{-1} P e^{i2\Lambda_> t'} P^{-1} Z.
\end{aligned}
\tag{B20}
$$

## Appendix C: Klich's determinant formula

Consider two second quantized operators $\hat{A}$ and $\hat{B}$, such that

$$
\hat{A} = \sum_{i,j} \langle i| A |j \rangle d_i^\dagger d_j,
$$

$$
\hat{B} = \sum_{i,j} \langle i| B |j \rangle d_i^\dagger d_j,
$$

where $A$ and $B$ are first the quantized operators and the states $|i\rangle$ span the corresponding single particle Hilbert space. Then it can be shown that [120]

$$
\mathrm{Tr}\left( e^{\hat{A}} e^{\hat{B}} \right) = \det \left( 1 - \xi e^A e^B \right)^{-\xi},
\tag{C1}
$$

where $\xi = 1$ for Bosons and $\xi = -1$ for Fermions (the creation and annihilation operators satisfy $d_i d_j^\dagger - \xi d_j^\dagger d_i = \delta_{ij}$).

We are going to derive a similar formula for operators that also contain terms of the form $d_i d_j$ and $d_i^\dagger d_j^\dagger$, we will work with Bosonic operators, the formula for Fermionic operators can be derived in a similar way.

Consider an operator (not necessarily Hermitian) that can be written in terms of Bosonic creation and annihilation operators as follows

$$
\hat{A} = \sum_{i,j=1}^J a_i^\dagger A_{ij}^{(0)} a_j + \frac{1}{2} a_i^\dagger A_{ij}^{(1)} a_j^\dagger + \frac{1}{2} a_i A_{ij}^{(2)} a_j = \frac{1}{2}\mathcal{A} - \frac{1}{2}\mathrm{tr}A^{(0)},
\tag{C2}
$$

(note: Tr is the trace for a many body operator while tr is the normal trace for a matrix), here

$$
\mathcal{A} = \mathbf{a}^\dagger A \mathbf{a},
$$

$$
A = \begin{pmatrix} A^{(0)} & A^{(1)} \\ A^{(2)} & A^{(0)T} \end{pmatrix}.
$$

The matrices $A^{(1)}$ and $A^{(2)}$ can always be written as symmetric matrices due to the Bosonic commutation relations. This leads to the following property

$$
XAX = A^T.
\tag{C3}
$$

To calculate the trace of this many body operator, we can diagonalize $ZA$ but since it is not Hermitian, there is no guarantee that it is diagonalizable. Instead we find Schur's decomposition of $ZA$ i.e.

$$
ZA = U\Lambda U^\dagger,
\tag{C4}
$$

where $U$ is a unitary matrix and $\Lambda$ is an upper triangular matrix with eigenvalues of $ZA$ on its diagonal. Schur's decomposition allows us to arrange the eigenvalues in any order we want on the diagonal of $\Lambda$, we will choose the following arrangement

$$
\mathrm{Diag}\Lambda = \begin{pmatrix} \lambda_1 & \cdots & \lambda_J & -\lambda_J & \cdots & -\lambda_1 \end{pmatrix},
\tag{C5}
$$

the above arrangement is possible because the eigenvalues of $ZA$ come in pairs of $(\lambda, -\lambda)$, this fact can be ascertained from the following property

$$\text{if } (ZA - \lambda)^m \, |\lambda\rangle = 0$$
$$\implies |\lambda\rangle^T \, ZX \, (ZA + \lambda)^m = 0, \tag{C6}$$

for $m \geq 1$, the $m \neq 1$ cases correspond to generalized eigenvectors. The specific arrangement of eigenvalues in eq.(C5) also imposes a specific structure on the matrix $U$, which is

$$U = \begin{pmatrix} |\lambda_1\rangle & \cdots & |\lambda_J\rangle & XZ|\lambda_J\rangle^* & \cdots & XZ|\lambda_1\rangle^* \end{pmatrix}, \tag{C7}$$

where $|\lambda_i\rangle$ is the (generalized) eigenvector corresponding to the eigenvalue $\lambda_i$. The eigenvectors are normalized as follows

$$|\lambda_i\rangle^\dagger \, |\lambda_j\rangle = \delta_{ij}. \tag{C8}$$

Now we return to the problem of calculating the trace of the many body operator, for that

$$\mathcal{A} = \mathbf{a}^\dagger A \mathbf{a}$$
$$= \mathbf{a}^\dagger Z U Z Z \Lambda U^\dagger \mathbf{a}$$
$$= \begin{pmatrix} c_1^\bullet & \cdots & c_J^\bullet & c_J & \cdots & c_1 \end{pmatrix} Z\Lambda \begin{pmatrix} c_1 & \cdots & c_J & c_J^\bullet & \cdots & c_1^\bullet \end{pmatrix}^T, \tag{C9}$$

and the commutation relations for the new operators are $[c_j, c_{j'}^\bullet] = \delta_{jj'}$, $[c_j, c_{j'}] = 0$, $[c_j^\bullet, c_{j'}^\bullet] = 0$. Finally,

$$\mathcal{A} = \sum_{i=1}^J \lambda_i \left( c_i^\bullet c_i + c_i c_i^\bullet \right) + \sum_{i,j=1}^J N_{ij} c_i^\bullet c_j^\bullet + \sum_{i<j,i=1}^J R_{ij} c_i^\bullet c_j, \tag{C10}$$

where the terms $N_{ij}$ and $R_{ij}$ encompass the non-diagonal terms of the upper triangular matrix $\Lambda$.

Define two states $\langle 0|_L$ and $|0\rangle_R$ such that

$$\langle 0|_L \, c_i^\bullet = 0 \; \forall i, \tag{C11}$$
$$c_i \, |0\rangle_R = 0 \; \forall i. \tag{C12}$$

Then it is easy to see that the operator $\sum_{i=1}^J c_i^\bullet c_i$ acts as a number operator. We can define a basis for a Fock space in terms of the following states

$$|n_1, n_2, \cdots, n_J\rangle = \frac{c_1^{\bullet n_1} c_2^{\bullet n_2} \cdots c_J^{\bullet n_J}}{\sqrt{n_1! n_2! \cdots n_J!}} \, |0\rangle_R, \tag{C13}$$

$$\langle\langle n_1, n_2, \cdots, n_J| = \langle 0|_L \frac{c_1^{n_1} c_2^{n_2} \cdots c_J^{n_J}}{\sqrt{n_1! n_2! \cdots n_J!}}. \tag{C14}$$

The states are normalized such that

$$\langle\langle n_1, n_2, \cdots, n_J| \, |m_1, m_2, \cdots, m_J\rangle = \delta_{n_1, m_1} \delta_{n_2, m_2} \cdots \delta_{n_J, m_J}, \tag{C15}$$

and the resolution of the identity on the Fock space is

$$\hat{\mathcal{I}} = \sum_{n_1=0}^\infty \cdots \sum_{n_J=0}^\infty |n_1, n_2, \cdots, n_J\rangle \, \langle\langle n_1, n_2, \cdots, n_J| \, . \tag{C16}$$

The trace of the exponential of the many body operator can now be written as

$$\text{Tr} e^{\hat{A}} = e^{-\text{tr} A^{(0)}/2} \text{Tr} e^{\mathcal{A}/2}$$
$$= e^{-\text{tr} A^{(0)}/2} \sum_{n_1=0}^\infty \cdots \sum_{n_J=0}^\infty \langle\langle n_1, n_2, \cdots, n_J| \, e^{\mathcal{A}/2} \, |n_1, n_2, \cdots, n_J\rangle$$
$$= \sum_{n_1=0}^\infty \cdots \sum_{n_J=0}^\infty \langle\langle n_1, n_2, \cdots, n_J| \, e^{\sum_{i=1}^J \lambda_i (c_i^\bullet c_i + c_i c_i^\bullet)/2} \, |n_1, n_2, \cdots, n_J\rangle$$
$$= \prod_{i=1}^J \frac{e^{\lambda_i/2}}{1 - e^{\lambda_i}}. \tag{C17}$$

The terms $\sum_{i,j=1}^{J} N_{ij} c_i^\bullet c_j^\bullet$ and $\sum_{i<j,i=1}^{J} R_{ij} c_i^\bullet c_j$ change the number states in such a way that no combination of them with each other or themselves can contribute to the trace, therefore the only term that contributes is $\sum_{i=1}^{J} \lambda_i (c_i^\bullet c_i + c_i c_i^\bullet)$. This was the reason why we chose a specific arrangement of eigenvalues in eq.(C5).

We can write eq.(C18) as

$$\mathrm{Tr}e^{\hat{A}} = e^{-\mathrm{tr}A^{(0)}/2} \det(1 - UU^\dagger Z e^{ZA/2})^{-1},\tag{C18}$$

which doesn't appear to be universal on account of the appearance of the matrix $U$, instead we will work with the following formula

$$\left[\mathrm{Tr}e^{\hat{A}}\right]^2 = \det(Z)e^{-\mathrm{tr}A^{(0)}} \det(1 - e^{ZA})^{-1}.\tag{C19}$$

We still don't have an expression which is analogous to the eq.(C1), to derive such an expression, consider another operator $\hat{B}$ that can be written in a form similar to eq.(C2), then it can be shown that

$$\left[\frac{1}{2}\mathcal{A}, \frac{1}{2}\mathcal{B}\right] = \frac{1}{2}\mathcal{C},\tag{C20}$$

where

$$\mathcal{C} = \mathbf{a}^\dagger Z \left[ZA, ZB\right] \mathbf{a}.$$

Using the Baker-Campbell-Hausdorff we can get our final expression as

$$\left[\mathrm{Tr}\left(e^{\hat{A}} e^{\hat{B}}\right)\right]^2 = \det(Z)e^{-\mathrm{tr}(A^{(0)}+B^{(0)})} \det(1 - e^{ZA} e^{ZB})^{-1}.\tag{C21}$$

## Appendix D: Correlation matrix from Klich's determinant formula

This section gives the expressions used to calculate the equal time correlation matrix after the quench $F_>(t,t)$, using the formulas developed in the last section. Before we give the general expressions, first let us note a result that will be used in the rest of the section

$$\lim_{\beta\to\infty} \left[\frac{\mathrm{Tr}\left(e^{\hat{A}} e^{-\beta\mathcal{H}_<}\right)}{\mathrm{Tr}\left(e^{-\beta\mathcal{H}_<}\right)}\right]^2 = \lim_{\beta\to\infty} e^{-\mathrm{tr}A^{(0)}} \frac{\det(1 - e^{ZA} e^{-\beta ZH_<})^{-1}}{\det(1 - e^{-\beta ZH_<})^{-1}}$$

$$= \lim_{\beta\to\infty} e^{-\mathrm{tr}A^{(0)}} \det\left(\frac{1}{1 - e^{-\beta\Lambda_<}} - Q^{-1} e^{ZA} Q \frac{e^{-\beta\Lambda_<}}{1 - e^{-\beta\Lambda_<}}\right)^{-1}$$

$$= e^{-\mathrm{tr}A^{(0)}} \det\left(\frac{\mathbb{I}_{2J} + Z}{2} + Q^{-1} e^{ZA} Q \frac{\mathbb{I}_{2J} - Z}{2}\right)^{-1},\tag{D1}$$

where the diagonal matrix $\Lambda_<$ of dimensions $2J$ has positive eigenvalues in the first half on the diagonal and negative eigenvalues in the second half.

Now let us write down the expressions for elements of the correlation matrix

$$F_{>,ij}(t,t) = \left\langle \mathbf{a}^\dagger(t)_i \mathbf{a}(t)_j \right\rangle = \partial_\chi \left\langle e^{\chi \mathbf{a}^\dagger(t)_i \mathbf{a}(t)_j} \right\rangle_{\chi=0}$$

$$\approx \lim_{\delta\chi\to 0} \frac{\left\langle e^{\delta\chi \mathbf{a}^\dagger(t)_i \mathbf{a}(t)_j} \right\rangle - 1}{\delta\chi}$$

$$= \lim_{\delta\chi\to 0} \frac{\sqrt{\left\langle e^{\delta\chi \mathbf{a}^\dagger(t)_i \mathbf{a}(t)_j} \right\rangle^2} - 1}{\delta\chi}$$

$$= \lim_{\delta\chi\to 0} \frac{\sqrt{\lim_{\beta\to\infty} \frac{\mathrm{Tr}\left(e^{i\mathcal{H}_> t} e^{\chi \hat{x}} e^{-i\mathcal{H}_> t} e^{-\beta\mathcal{H}_<}\right)^2}{\mathrm{Tr}\left(e^{-\beta\mathcal{H}_<}\right)^2}} - 1}{\delta\chi},\tag{D2}$$

where we have defined the operator $\hat{\chi}$ as follows

$$\hat{\chi} = \delta\chi \left( \frac{1}{2}\mathbf{a}^\dagger M \mathbf{a} \pm \frac{1}{2}\mathrm{tr}M^{(0)} \right),$$

$$M = \begin{pmatrix} M^{(0)} & M^{(1)} \\ M^{(2)} & \left(M^{(0)}\right)^T \end{pmatrix}, \tag{D3}$$

such that

$$M^{(1)} = \left(M^{(1)}\right)^T,$$

$$M^{(2)} = \left(M^{(2)}\right)^T.$$

The elements of the matrix $M$ and the plus or minus sign in the definition of $\hat{\chi}$ depend on the value of the indices $i$ and $j$.

$$F_{>,ij}(t,t) = \lim_{\delta\chi \to 0} \frac{\sqrt{\lim_{\beta \to \infty} \frac{\mathrm{Tr}\left(e^{i\mathcal{H}_>t}e\hat{\chi}e^{-i\mathcal{H}_>t}e^{-\beta\mathcal{H}_<}\right)^2}{\mathrm{Tr}\left(e^{-\beta\mathcal{H}_<}\right)^2} - 1}}{\delta\chi}$$

$$= \lim_{\delta\chi \to 0} \frac{\sqrt{\lim_{\beta \to \infty} \frac{e^{\pm\delta\chi \mathrm{tr}M^{(0)}}\det\left(1-e^{iZ_{2J}H_>t}e^{\delta\chi Z_{2J}M}e^{-iZ_{2J}H_>t}e^{-\beta Z_{2J}H_<}\right)^{-1}}{\det\left(1-e^{-\beta Z_{2J}H_<}\right)^{-1}} - 1}}{\delta\chi}, \tag{D4}$$

$$\tag{D5}$$

where we have used Eq.(C21). Expanding the exponentials upto first order in $\delta\chi$, and using the identity $\det(\mathbb{I} + \epsilon A) \approx 1 + \epsilon \mathrm{tr}(A) + \mathcal{O}(\epsilon^2)$, we get

$$F_{>,ij}(t,t) = \lim_{\delta\chi \to 0} \frac{e^{\pm\delta\chi \mathrm{tr}\frac{M^{(0)}}{2}}\left(1 + \frac{\delta\chi}{2}\mathrm{tr}\left[Q^{-1}e^{iZH_>t}ZMe^{-iZH_>t}Q\left(\mathbb{I}_{2J} - Z\right)\right]\right)^{-1/2} - 1}{\delta\chi}. \tag{D6}$$

If $\mathrm{tr}M^{(0)} = 0$

$$F_{>,ij}(t,t) = \lim_{\delta\chi \to 0} \frac{\left(1 + \frac{\delta\chi}{2}\mathrm{tr}\left[Q^{-1}e^{iZH_>t}ZMe^{-iZH_>t}Q\left(\mathbb{I}_{2J} - Z\right)\right]\right)^{-1/2} - 1}{\delta\chi}$$

$$\approx \lim_{\delta\chi \to 0} \frac{1 - \frac{\delta\chi}{4}\mathrm{tr}\left[Q^{-1}e^{iZH_>t}ZMe^{-iZH_>t}Q\left(\mathbb{I}_{2J} - Z\right)\right] - 1}{\delta\chi}$$

$$= -\frac{\mathrm{tr}\left[Q^{-1}e^{iZH_>t}ZMe^{-iZH_>t}Q\left(\mathbb{I}_{2J} - Z\right)\right]}{4}. \tag{D7}$$

instead if $\mathrm{tr}M^{(0)} = 1$ with a negative sign in the definition of $\hat{\chi}$

$$F_{>,ij}(t,t) = \lim_{\delta\chi \to 0} \frac{e^{-\frac{\delta\chi}{2}}\left(1 + \frac{\delta\chi}{2}\mathrm{tr}\left[Q^{-1}e^{iZH_>t}ZMe^{-iZH_>t}Q\left(\mathbb{I}_{2J} - Z\right)\right]\right)^{-1/2} - 1}{\delta\chi}$$

$$\approx \lim_{\delta\chi \to 0} \frac{e^{-\frac{\delta\chi}{2}} - e^{-\frac{\delta\chi}{2}}\frac{\delta\chi}{4}\mathrm{tr}\left[Q^{-1}e^{iZH_>t}ZMe^{-iZH_>t}Q\left(\mathbb{I}_{2J} - Z\right)\right] - 1}{\delta\chi}$$

$$= -\frac{1}{2} - \frac{\mathrm{tr}\left[Q^{-1}e^{iZH_>t}ZMe^{-iZH_>t}Q\left(\mathbb{I}_{2J} - Z\right)\right]}{4}. \tag{D8}$$

and finally if $\mathrm{tr}M^{(0)} = 1$ with a plus sign in the definition of $\hat{\chi}$

$$F_{>,ij}(t,t) = \lim_{\delta\chi \to 0} \frac{e^{\frac{\delta\chi}{2}}\left(1 + \frac{\delta\chi}{2}\mathrm{tr}\left[Q^{-1}e^{iZH_>t}ZMe^{-iZH_>t}Q\left(\mathbb{I}_{2J} - Z\right)\right]\right)^{-1/2} - 1}{\delta\chi}$$

$$\approx \lim_{\delta\chi \to 0} \frac{e^{\frac{\delta\chi}{2}} - e^{\frac{\delta\chi}{2}}\frac{\delta\chi}{4}\mathrm{tr}\left[Q^{-1}e^{iZH_>t}ZMe^{-iZH_>t}Q\left(\mathbb{I}_{2J} - Z\right)\right] - 1}{\delta\chi}$$

$$= \frac{1}{2} - \frac{\mathrm{tr}\left[Q^{-1}e^{iZH_>t}ZMe^{-iZH_>t}Q\left(\mathbb{I}_{2J} - Z\right)\right]}{4}. \tag{D9}$$

The matrix $M$ and the operator $\hat{\chi}$ will have the following forms depending on the indices $i$ and $j$.

- $i \leq J, j \leq J$:

  The matrices $M^{(1)}$ and $M^{(2)}$ will only contain zeros, and

$$M_{lk}^{(0)} = \begin{cases} 0 & l \neq i, k \neq j, \\ 1 & l = i, k = j, \end{cases}$$

$$\hat{\chi} = \delta\chi \left( \frac{1}{2}\mathbf{a}^\dagger M \mathbf{a} - \frac{1}{2}\mathrm{tr}M^{(0)} \right), \tag{D10}$$

  the indices $l$ and $k$ run from 1 to $J$. Here, $\mathrm{tr}M^{(0)} = 1$ only if $i = j$ otherwise it's zero.

- $i \leq J, j > J$:

  The matrices $M^{(0)}$ and $M^{(2)}$ will only contain zeros, and

$$M_{lk}^{(1)} = \begin{cases} 1 & l = j - J, k = i, \\ 1 & l = i, k = j - J, \\ 0 & \text{every other element,} \end{cases}$$

$$\hat{\chi} = \delta\chi \frac{1}{2}\mathbf{a}^\dagger M \mathbf{a}. \tag{D11}$$

  Here, $\mathrm{tr}M^{(0)}$ is always zero.

- $i > J, j \leq J$:

  The matrices $M^{(0)}$ and $M^{(1)}$ will only contain zeros, and

$$M_{lk}^{(2)} = \begin{cases} 1 & l = j, k = i - J, \\ 1 & l = i - J, k = j, \\ 0 & \text{every other element,} \end{cases}$$

$$\hat{\chi} = \delta\chi \frac{1}{2}\mathbf{a}^\dagger M \mathbf{a}. \tag{D12}$$

  Again, $\mathrm{tr}M^{(0)}$ is always zero.

- $i > J, j > J$:

  The matrices $M^{(1)}$ and $M^{(2)}$ will only contain zeros, and

$$M_{lk}^{(0)} = \begin{cases} 0 & l \neq i - J, k \neq j - J, \\ 1 & l = i - J, k = j - J, \end{cases}$$

$$\hat{\chi} = \delta\chi \left( \frac{1}{2}\mathbf{a}^\dagger M \mathbf{a} + \frac{1}{2}\mathrm{tr}M^{(0)} \right). \tag{D13}$$

  Here, $\mathrm{tr}M^{(0)} = 1$ only if $i = j$ otherwise it's zero.

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
