# Peer review of "On-demand analog space-time in superconducting networks: grey holes, dynamical instability and exceptional points"

_SciPost Physics_

## Round 4 · Referee Report · Anonymous (Referee 1) · 2025-10-25

Report

The work by Javed et al. proposes an interesting approach to probe some of the physics of black holes in a superconducting circuit. The authors introduce various circuit models comprising some standard circuit elements in cQED and some less standard elements such as gyrators and negative inductances, that add the necessary ingredients to emulate black hole behavior.
I think generally the paper reads nicely and seems scientifically sound. I also find that idea introduced, while possibly rather far from experiments, is intriguing and novel enough to deserve publication.

I have a few questions that I believe could strengthen the paper if addressed.

1) Gyrators introduce a magnetic field in cQED, breaking time-reversal symmetry. It is not clear to me whether one generally needs time-reversal symmetry breaking to simulate black holes. If black holes do not require T-symmetry breaking, while the device that simulates them does, could some aspects of the underlying physics be lost or altered in the simulation?

2) In the device modelled, if the gyrator adds a magnetic field, would this not lead to “Peierls phases” to be accumulated from one unit cell to the next? In this case, I would expect some kind of quantisation condition for the allowed values of G when a periodic system is considered. Could the authors comment on whether such constraints arise in their model, and if so, how they are handled?

3) Since the authors seem to be focusing on a lumped element implementation of the device, it could be helpful to comment on the effect of disorder, especially in terms of variability of the circuit parameters. Could the authors comment on which parameters are most critical to keep uniform for the analogy to hold? A short discussion (or numerical estimate) of the sensitivity of the model to disorder would help guide future efforts.

Recommendation

Publish (meets expectations and criteria for this Journal)

  • validity: -
  • significance: -
  • originality: -
  • clarity: -
  • formatting: -
  • grammar: -

Author:  Mohammad Atif Javed  on 2025-12-19  [id 6158]

(in reply to Report 1 on 2025-10-25)
Category:
answer to question

We warmly thank the referee for the favorable report and finding our ideas intriguing and novel. We believe that our revision of the manuscript in response to the referee’s comments have allowed us to increase the depth of our work.

The answer to the referee's questions are formulated below, for a detailed overview of the changes made to the paper please see the "List of changes" to be submitted with the revised manuscript.

1) Considering simulations of apparent horizons as originally proposed by Unruh (see also the review by Robertson, Ref. (43)) breaking time-reversal symmetry (TRS) is necessary. Typically, the goal is to overtilt the dispersion relation making both branches travel in the same direction. Any non-zero tilt (i.e. non-zero v in Eq. (2)) invokes terms that break time-reversal symmetry, specifically the terms linear in time derivatives of the scalar field. While other propositions within circuit QED (mostly the works dealing with solitons Ref. (33)) do not seem to require explicit TRS breaking elements, the emergent effective Lagrangian still violates TRS as it is formulated in the co-moving soliton frame. Our work therefore differs from previous works by incorporating TRS breaking via an explicit circuit element. This gives us a better control of the effective spacetime geometry, allowing us to study various unconventional features such as: grey holes and their connection to exceptional points. In the manuscript we have added a corresponding footnote (Footnote 1).

2) We thank the referee for this intriguing question. The referee is correct with their expectation that G takes on discrete values, when considering compact (2 periodic) superconducting phase (ϕ) values for each node. This is simply the torus version of Dirac’s well-known magnetic monopole quantization argument. We are acutely aware of this feature, and some of us have already embarked on an extensive research endeavor exploring a number of fascinating consequences due to quantized gyration conductance (see Ref. (72)). However, in order for the quantization of G and the compactness of ϕ to become relevant, the system at low energies needs to be able to explore the full ϕ space. We consider a regime where the Josephson energies are high enough such that phase differences remain in the troughs of the cosine potential. In that case the value of G does not necessarily correspond to the Chern number defined as the integral of the Berry curvature over the entire (2 periodic) ϕ space, but instead corresponds to local expansion of the Berry curvature close to the energy minima. This local Berry curvature does not need to yield quantized values, and in addition can continuously fluctuate from one gyrator element to the next. This motivated us to include disorder in the gyration value, also in response to the referee’s third question. In case the referee is interested in learning about the detailed consequences in the opposite regime of weak Josephson effects, we recommend considering our previously mentioned work on the topic, where quantized G (even inhomogeneous G values) are considered and predicted to give rise to anyonic excitations with strange non-local features. We have added a footnote (Footnote 3) to this effect.

3) We have added the Appendix F and additional text at the end of Sec. 5, briefly discussing the effect of disorder in the gyrator and inductive elements. The main result of this investigation is that for relative disorder strength of 10% and below, the evaporation of horizons in both the circuits is insignificantly affected. For the same disorder the quantum fluctuations in the wormhole region remain unchanged for the circuit with only nearest neighbor connections but show some deviation from the case of no disorder for the circuit where JJs connect next nearest neighbors. We also find that for similar relative strength of disorder, the disorder in gyrator elements is more impactful than the disorder in inductive elements. This can be understood by looking at Eq. (20) in the manuscript combined with the fact that the parameter regime we work in requires 4G^{2}>E_{L}/E_{C}, this leads to any change in gyrator conductance being more effective than a change in inductance.

---

## Round 4 · Referee Report · Anonymous (Referee 2) · 2025-10-29

Disclosure of Generative AI use

The referee discloses that the following generative AI tools have been used in the preparation of this report:

Large language models (ChatGPT, Deepseek) to check if some of my comments were reasonable, and to ask for introductory/summarizing material about some fields I am not well versed in.

Strengths

1- Compelling motivation for the use of superconducting circuits as analog spacetime geometries. The possibilities seem richer than the actual gravitational metrics.

2- Explicit solutions for the modes in novel 1D array configurations, with Josephson junctions and circulators. This has interest well beyond the application as a simulator.

Weaknesses

1- It is not very clear how several of the analogue gravity issues that are thoroughly discussed in the Introduction manifest in the explicit calculations performed later.

2- The proposals require circuit elements, such as gyrators and negative inductances, that are still in early stages of development.

Report

This work proposes the use of novel superconducting circuits as black hole analogs by emulating different spacetime geometries in a controlled manner. Along the way, it provides interesting results, such as calculations for the modes in 1D arrays with gyrators and the development of some technical tools. These result are timely, since the implementation of some of these novel circuit elements is in progress [though this also hinders the applicability of the proposals], and provide a synergetic link with spacetime geometry. However, the manuscript is long and involved, and could benefit from a clarification effort on some issues:

1- The introduction section highlights several potential advantages of the gyrator arrays with respect to other implementations. Could the authors provide a more detailed comparison between platforms [not necessarily in this chapter]? E.g., between the "control over the shape of the space-time geometry" used in convential Josephson junction arrays and the flux quenches. Another related platform is the electromagnetic waveguide of Ref. 46, which just uses (time-dependent) linear elements. Also, in the same work an important issue about the need of quantum effects for Hawking radiation is pointed out [cf. Ref. 9 therein]. Such a discussion, adapted to the proposals of this article, would benefit the comprehensiveness of the introduction.

2- Though not with detail, the authors propose the use of tunable 0-$\pi$ junctions to shift the Josephson potential. However, it shall be noted that typically, in a 0-$\pi$ transition, the ground state changes parity, so the initial and final potentials would not be coherently connected - an external quasiparticle is required to change between such states. Also, could a similar $\phi\sim\pi$ shift be generated by substituting the single junction with an assymmetric SQUID? [even if the effective EJ would change].

3- The authors may specify in which context and in which degree of generality the work "provides a first step towards investigating the backaction of quantum fields on spacetime". The <qualitative speculations> of Sec. 6 seem to indicate that the geometry is dictated/interpreted from a convenient choice that depends on the wavefunctions (e.g. the quadratic expansions of the cosine).

4- Given the importance of the full $\cos \phi$ potential, what is to be expected from the interactions produced by an expansion higher than quadratic?

5- The introductory remarks of Sec. 5 would benefit from a further motivation for the choice of two point correlation function analysis. This could include, e.g., reminding/deepening the reasons given in Sec. 2.2 or a (brief) summary of the methods used in the literature to identify non-trivial geometries and radiation - perhaps moving here part of the last paragraph of Sec. 4.

6- What is the motivation for the choice of the parameters in Figs. 6 and 7? What determines the frequency of the modulation in Fig. 7a? It is a bit surprising that, after such a challenging implementation of the calculations, there is not a more detailed exploration of the system. The comparison between the methods could also be more explicit in Sec. 5 (not only in Sec. 7). Also, while in the bands the EP may pose problems to the direct diagonalization method, in the finite size system it may be that the eigenvalues are not defective. And even if this is the case, why does it prevent to form a complete basis able to describe the dynamics?

Requested changes

The authors may consider the questions posed in the Report, and other minor comments/suggestions in (page number):
(1) "combining" the "nonreciprocity [] with the nonlinearity"
(5) "it is understood since a long time" requires a reference, perhaps a review; "looses"->"loses" ¿?; can you write the transform back to (t,x) explicitly (for constant u,v)?
(15) Fig. 4. yellow and orange colors are very similar. Vertical lines in $\pm \pi$ could help (also in Fig. 3).
(18) Fig. 5. Label "b)"
(20) "black and white hole horizon"s
(23) "three types", perhaps needs to be more explicit
(27) Eq. A.7. $\mathcal{H}_{1,m}$ ¿?
(30) $2J*2J$ and other instances, perhaps $2J\times 2J$ is more common
(40) Fig. 9a. Labels for $\alpha=0.5, 0.9$ are too similar. Why not draw the theory line for all of them and avoid panel b?
Refs.- Some topics discussed in the article could benefit from a broadened literature research, such as the quantization of nonreciprocal elements or the 0-$\pi$ junctions.

Recommendation

Ask for major revision

  • validity: good
  • significance: good
  • originality: high
  • clarity: ok
  • formatting: reasonable
  • grammar: good

Author:  Mohammad Atif Javed  on 2025-12-19  [id 6159]

(in reply to Report 2 on 2025-10-29)

We thank the referee for the positive assessment and careful reading of the manuscript. We believe that in responding to the referee’s various very justified questions and suggestions, our paper has been significantly improved.

The answers to referee's questions can be found below, for an overview of the changes made to the paper please see the "List of changes" to be submitted with the revised manuscript.

1) In other proposals for simulating spacetime geometry in circuit QED [32,33,46], to recreate a spacetime with anisotropy in signal speeds but no horizon, already requires a time dependent electromagnetic field that sweeps through the circuit. This leads to the issue that even a stationary spacetime (i.e. spacetime with time translation symmetry), such as a spacetime with no apparent horizons, requires a time dependent circuit which is not the case when using gyrators. Additionally, when investigating a non-stationary spacetime, such as one with an evaporating horizon where we do use a time-dependent flux, using gyrators has the advantage of fixing the position of the horizons in the lab frame as opposed to a co-moving frame that moves with the sweeping electromagnetic field. We have added a remark to this effect in Section 2.2, on Page 8.

2) We thank the referee for pointing out this caveat about fermion parity. We do not expect this aspect to be critical for our proposal, as different fermion parity sectors can be incorporated in circuit QED through periodic and anti-periodic boundary conditions in ϕ space. Additionally, the boundary conditions do not play a role in the here considered regime of large Josephson energies. Furthermore, our proposal does not necessitate coherent superpositions between the 0 and \Pi configurations, such that we do not expect any conflict with the Wigner superselection rule. The main bottleneck requirement is a sufficiently fast swap from 0 to \Pi configuration, such that the many-body wavefunction is located at the maximum of the cosine immediately after the quench.

Finally, the referee is correct in expecting that an asymmetric SQUID works as well in implementing a \Pi shift. It has to be noted though that the mechanism is exactly the same as for a single Josephson junction, and moreover in a generic SQUID device the efficiency of the conversion from the current amplitude to the phase shift is reduced. This can be easily seen when considering the SQUID Josephson energy as a sum of two shifted cosines, where the external current couples with a likewise asymmetric coupling prefactor to both the strong and the weak junction. The effective shift of the total Josephson energy depends on the asymmetry of this coupling. The efficiency is maximal if the current couples exclusively to the larger of the two junctions and minimal when it couples to the smaller junction. We have added a small calculation (paragraph surrounding Eq. (20)) showing this on Page 20.

3) The answer to this question is strongly related to our answer to referee’s next question. Please consider our answer below to point 4.

4) The full cos(ϕ) potential plays a special role in a limited sense, we need it in order to induce a \Pi shift in the potential and create negative inductances. However we consider a regime of large Josephson energies w.r.t. other energy scales, i.e. before and immediately after the quench the many-body wavefunction remains localized justifying the quadratic expansion of the cosine. The non-linearity of the cosine will without a doubt play a crucial role beyond transient times, as also discussed in Sec. 6 of the manuscript. We sincerely hope the referee understands that this is an extremely complex question that requires an entirely dedicated new research effort. For instance non-linear corrections will give rise to interacting bosons and potentially quantum phase-slip physics. We point to that possibilities in an additional phrase in the outlook.

Based on this we can also answer referee’s question in point 3 above. The referee correctly understood that our approach relies on taking a reference point in the Hilbert space spanned by the superconducting phases and expanding around it. In the wormhole regime, this expansion is by definition unstable, due to exponential accumulation of quantum fluctuations, which cause problems beyond transient times, due to non-linearity. As we pointed out above in our response, the full solution to this interacting (non-linear) problem is hard to obtain. But including dissipation, which can qualitatively be captured by fuzzy projections in the space of superconducting phases, the problem can be simplified again, expanding around different minima (after environment-induced projection), leading to a (time-dependent) diffusion of spacetime, and thus inevitably to backaction, since the expanded (nearly linear) scalar field always reacts to the updated spacetime configuration. In fact, in this regard, our expansion approach might actually not only serve as a simulation platform, but conversely, the diffusive spacetime perspective could also serve as a useful conceptual trick to capture strongly non-linear systems in the presence of dissipation. We have added a remark to this end in the outlook.

With the unique use of non-linearity specific to our work, and with the potential connection to backaction on space-time geometry, we believe that our work highlights an important, and previously disregarded regime of non-linear devices prepared in a precarious unstable starting state, suitable for multi-pronged future research efforts.

5) We thank the referee for this remark to improve the motivation for using two-point correlation functions. We chose to follow their second suggestion and refer to previous literature using two-point correlation functions to examine features related to analog event horizons. The beginning of Section 5 now contains an additional text passage referring specifically to experiments done in cold atoms (predominantly Steinhauer’s work), where two-point correlation functions were central to probe the thermal nature of analog Hawking radiation.

6) We have added a more detailed discussion of dispersion relation in Eq. (20) on Page 14, pointing out that there are only two dimensionless free parameters, the gyration conductance G and the ratio of the inductive energy to the charging energy E_L/E_C. While the latter has to be larger than 1 to justify the harmonic approximation of the cosine, there is per se no restriction on G (up to a quantization constraint due to phase compactness, see also other referee report). However, the relative magnitude of E_L/E_C and G is not of importance for the creation of analog event horizons – the only thing that matters is the sign of E_L/E_C. The choice of parameters in Figs. 6 and 7 merely reflects this observation.

The frequency of modulation in Fig. 7a is a consequence of the quench and is independent of the presence of the horizons, hence this feature is still present when the quench is performed in the entire circuit (i.e. no horizons). We expand on this in added text on Page 23 and in the newly added Fig. 8.

The existence of EPs in the parameter space of the system, boils down to checking if the relevant matrix is diagonalizable or not, for every value of system parameters. The check if a given matrix is diagonalizable, apart from special cases where the spectral theorem is applicable, is in general numerically expensive (for large matrices) and not very accurate if the eigenvalues are very close to each other. Hence, it is better to bypass this procedure and not worry about diagonalizing a possibly defective (non-diagonalizable) matrix. Additionally, it follows from the definition of diagonalizability that a defective matrix cannot have a complete basis of eigenvectors. One can, in principle, work with generalized eigenvectors, but this again brings us back to the problem of first determining whether a given matrix is defective or not. We have added additional text in Section 5 explaining the motivation for using Klich’s determinant formula.

---

## Round 5 · Author Response

Dear Editor,

We would like to thank you for the opportunity to revise our paper after the first round of review, we also thank the referees for their constructive comments on our work. We believe that our responses to their questions and the modifications made to the manuscript will be to their satisfaction.

The first referee had an overall positive assessment of our work and recommended it for publication. The referee had additional questions regarding some aspects of our work, such as the effect of breaking time reversal symmetry, quantization of gyrator conductance, and the effect of disorder. We have submitted our reply to the referee and modified our manuscript to include a discussion on these topics.

The second referee also has an overall positive assessment of our work, but suggested some revisions before recommending it for publication. The referee had questions regarding the novelty of using gyrators to induce a tilt in the dispersion relation, potential of using SQUIDs instead of Josephson junctions in the circuit, how our work provides a first step towards investigating the backaction of quantum fields on spacetime, the inclusion of higher order terms in the expansion of the cosine potential, the reason for using two-point correlation functions, and our choice of system parameters for the numerics. We have submitted our reply to the referee and modified our manuscript to include a discussion on these topics with additional figures and equations.

The complete list of changes made to the paper has also been submitted. We believe that the revised manuscript is now suitable for publication in SciPost Physics Core.

Sincerely,
The authors.

---

## Round 5 · List of Changes

List of major changes made to the manuscript:

1) Included additional text with citations in the second paragraph on Page 5 to support the claim that a variety of non-relativistic systems are described by Equation (1). The relevant text is “However, it is understood …… waves in Josephson junction arrays.”

2) Included a footnote on Page 5 (footnote 1) discussing the necessity of breaking time reversal symmetry (TRS) in order to simulate event horizons in condensed matter systems.

3) Added text in the first paragraph on Page 8 regarding the advantage of using gyrators, in circuit QED systems, to induce a tilt in the dispersion relation. The text is “This anisotropy in the group velocities ……. co-moving frame that moves with the sweeping electromagnetic field.”

4) Included a footnote on Page 13 (footnote 3) discussing the reason we can ignore the quantization of gyrator conductance (G) due to compactness of the phase variable in this work.

5) Added text in the last paragraph on Page 14 motivating the choice of system parameters to perform numerics in Section 5. The text is “Therefore, the central object …….. creating an overtilt. ”

6) Changed the color of the imaginary part of eigenvalues in Figure (4a) from yellow to green.

7) Added text in the third paragraph on Page (20), including Equation (25), discussing the potential of using a SQUID instead of a JJ to induce a phase shift of \Pi in order to achieve negative inductance for transient times. The text is “Alternatively ….. flux control hardware.”

8) Added text in the fourth paragraph on Page 21, motivating the use of two-point correlation functions to study analog event horizons. The text is “Additionally, ….. the Hawking temperature.”

9) Added text in the third paragraph on Page 22, motivating the use of Klich’s determinant formula instead of diagonalizing a possibly defective matrix. The text is “(non-diagonalizable) ……. a defective matrix.”

10) Added text in the third paragraph on Page 23 including Equation (29), explaining the presence of oscillatory behavior observed in Figure (7a) and providing the expression for the modulating frequency.

11) Added Figure (8) to support the text mentioned in the bullet point 10.

12) Added discussion at the end of Section 5 (beginning on Page 24 “Lastly, we also …….”) regarding the effect of disorder in gyrators and inductors on the time evolution of the two-point correlation functions after the quench.

13) Added a new appendix (Appendix F) with detailed numerics to investigate the effect of disorder, as mentioned in bullet point 12, along with new figures (Figures 12, 13, 14 and 15).

14) Added text in Section 7 on Page 28 as an outlook, discussing the possible effect of using higher order terms in the expansion of the cosine potential, and the possibility of studying the backaction of quantum fields on spacetime geometry in circuit QED. The text is “This work also …….. suitable for follow-up research.”

15) Removed the Figure (9) (in the old manuscript) in Appendix E, and replaced it with two new ones Figures (10) and (11).

---

## Editorial Decision

in_refereeing